# RpoS is a pleiotropic regulator of motility, biofilm formation, exoenzymes, siderophore and prodigiosin production, and trade-off during prolonged stationary phase in *Serratia marcescens*

**Han Qin, Ying Liu, Xiyue Cao, Jia Jiang, Weishao Lian, Dairong Qiao, Hui Xu\*, Yi Cao ⃝\***

Microbiology and Metabolic Engineering of Key Laboratory of Sichuan Province, College of Life Science, Sichuan University, Chengdu, P.R. China

\* geneium@scu.edu.cn (YC); xuhui_scu@scu.edu.cn (HX)

## Abstract

Prodigiosin is an important secondary metabolite produced by *Serratia marcescens*. It can help strains resist stresses from other microorganisms and environmental factors to achieve self-preservation. Prodigiosin is also a promising secondary metabolite due to its pharmacological characteristics. However, pigmentless *S. marcescens* mutants always emerge after prolonged starvation, which might be a way for the bacteria to adapt to starvation conditions, but it could be a major problem in the industrial application of *S. marcescens*. To identify the molecular mechanisms of loss of prodigiosin production, two mutants were isolated after 16 days of prolonged incubation of wild-type (WT) *S. marcescens* 1912768R; one mutant (named 1912768WR) exhibited reduced production of prodigiosin, and a second mutant (named 1912768W) was totally defective. Comparative genomic analysis revealed that the two mutants had either mutations or deletions in *rpoS*. Knockout of *rpoS* in *S. marcescens* 1912768R had pleiotropic effects. Complementation of *rpoS* in the Δ*rpoS* mutant further confirmed that RpoS was a positive regulator of prodigiosin production and that its regulatory role in prodigiosin biosynthesis was opposite that in *Serratia* sp. ATCC 39006, which had a different type of *pig* cluster; further, *rpoS* from *Serratia* sp. ATCC 39006 and other strains complemented the prodigiosin defect of the Δ*rpoS* mutant, suggesting that the *pig* promoters are more important than the genes in the regulation of prodigiosin production. Deletion of *rpoS* strongly impaired the resistance of *S. marcescens* to stresses but increased membrane permeability for nutritional competence; competition assays in rich and minimum media showed that the Δ*rpoS* mutant outcompeted its isogenic WT strain. All these data support the idea that RpoS is pleiotropic and that the loss of prodigiosin biosynthesis in *S. marcescens* 1912768R during prolonged incubation is due to a mutation in *rpoS*, which appears to be a self-preservation and nutritional competence (SPANC) trade-off.

**Data Availability Statement:** The complete genome of Serratia marcescens 1912768R was deposited in GenBank under the accession number

CP040350. The genomes of Serratia marcescens 1912768WR and 1912768W were deposited in GenBank under the accession number VBRI00000000 and VBRJ00000000, respectively.

**Funding:** This work was supported by National Natural Science Foundation of China (31670078); Sichuan Science and Technology Program (2018GZ0375 and 2018TJPT0004); Chengdu Science and Technology Program (2017-GH02-00071-HZ and 2018-YF05-00738-SN); National Infrastructure of Natural Resources for Science and Technology Program of China (NIMR-2018-8); Science and Technology Program of Sichuan University (2018SCUH0072).

**Competing interests:** The authors have read the journal's policy and the authors of this manuscript have the following competing interests to declare: The Shanghai Majorbio Bio-Pharm Technology Co., Ltd. performed the de novo genomic sequencing of Serratia marcescens 1912768R and sequencing of Serratia marcescens 1912768WR and 1912768W. This does not alter our adherence to PLOS ONE policies on sharing data and materials. There are no patents, products in development or marketed products to declare.

## Introduction

*Serratia marcescens*, a gram-negative bacterium belonging to the *Enterobacteriaceae* family, is found in a wide range of ecological niches in the environment [1]. It is known for its ability to synthesize the striking red pigment prodigiosin, which is of great interest for its antimicrobial as well as immunosuppressant and other bioactive properties [2–4]. *S. marcescens* is a model strain for studying the regulation of prodigiosin production and other secondary metabolites, such as carbapenem and serratamolide [5–7]. The biosurfactant serratamolide, which is called serrawettin W1 and catalyzed by the nonribosomal peptide synthetase SwrW, has antimicrobial and hemolytic properties [8, 9]. *S. marcescens* is also known to secrete a variety of exoenzymes, such as pectate lyase, lipase and cellulase [10].

Prodigiosin is synthesized by the prodigiosin biosynthesis gene cluster designated *pig* [6]. The *pig* clusters of 34 strains were divided into three types [11]: type I *pig* clusters are flanked by *cueR* and *copA* and shared by 29 strains; in type II *pig* clusters, which have been found in 4 strains, *cueR* and *copA* are downstream of the *pig* cluster and adjacent to each other; in type III *pig* clusters, which have been identified in *Serratia* sp. ATCC 39006, *pigO* is present downstream of the *pig* cluster. The biosynthesis of prodigiosin is under tight control by factors including Hfq, RpoS, cAMP receptor protein (CRP), EepR, HexS, LuxS, RssAB, SmaI and SpnR [7, 12–18]. The stationary phase sigma factor *rpoS* or $\sigma^s$ is a transcriptional factor and considered to be a global regulator that controls approximately 23% of the genes in the *E. coli* genome [19]. Polymorphisms of RpoS are extremely common in *E. coli* [19]. In *E. coli* and *Salmonella enterica* serovar Typhi, RpoS regulates resistance to stresses such as high osmolarity, oxygen free radicals, acid and other biotic stresses [20–22]. Mutation of *rpoS* reduces stress resistance but promotes nutritional competence, as bacteria allocate the limited resources to growth rather than self-protection when suffering starvation and vice versa; therefore, there is a self-preservation and nutritional competence (SPANC) trade-off in *E. coli* [23]. However, the effect of RpoS on the stress resistance and fitness of *S. marcescens* is unknown and remains to be explored.

In the natural environment, both pigmented and nonpigmented *S. marcescens* strains exist [24]; moreover, nonpigmented *S. marcescens* mutants can be easily isolated from WT pigmented strains after prolonged starvation [25]. However, the underlying mechanisms causing the loss of prodigiosin production in prolonged starving environments remain incompletely understood. A particular laboratory model for approximating prolonged starving environments is prolonged stationary phase [26]. Bacteria suffer a range of stresses during prolonged stationary phase in which starvation is the major challenge [27]. In *Escherichia coli*, a wide range of mutations accumulate in prolonged stationary phase over time due to the selection of the environment [28–30]; spontaneous mutants that confer a growth advantage in stationary phase (GASP) are enriched which is an adaptive evolution by natural selection [31]. The tight regulatory networks and flexible genetic characteristics of *E. coli* contribute to the survival of the bacteria during prolonged stationary phase [31–33]. Gene regulation provides adequate but limited control over the right level of gene expression in a less stressful environment, but mutational changes offer intensive control for the adaption of bacteria in prolonged stationary phase [27].

In this study, the pigmented WT strain *S. marcescens* 1912768R was isolated from rhizosphere soil in ginger fields and we also found that mutants defective in prodigiosin production were consistently isolated during prolonged stationary phase. In order to characterize the underlying molecular mechanism of the loss of prodigiosin production, comparative genomic profiling was employed between the WT strain and its prodigiosin defective mutants isolated during prolonged stationary phase. From these analyses, *rpoS* was identified as a candidate

transcriptional regulatory gene causing the loss of prodigiosin biosynthesis. The regulatory role of RpoS in prodigiosin production and other phenotypes was further studied. Additionally, the effects of the mutation in *rpoS* on the competition of *S. marcescens* 1912768R during prolonged stationary phase were explored.

## Materials and methods

### Bacterial strains, plasmids and growth conditions

*S. marcescens* 1912768R was isolated from rhizosphere soil in ginger fields and collected at the China Center for Type Culture Collection (strain number: CCTCC AB 2019405). Prodigiosin-defective mutants 1912768WR and 1912768W were randomly selected after 16 days of prolonged culture in Lysogeny Broth (LB; 10 g l$^{-1}$ tryptone, 5 g l$^{-1}$ yeast extract, and 10 g l$^{-1}$ NaCl) medium without any nutrition supplementation and isolated on LB agar plate by the pink and white color of the bacterial colony, respectively.

Bacterial strains and plasmids used in this study are listed in Table 1. *S. marcescens* strains were grown at 28°C and *E. coli* strains were grown at 37°C in LB medium or on solid LB agar plate (LBA) supplemented with 2.0% (w/v) agar. Bacterial growth was measured in a spectrophotometer and expressed as $OD_{600}$. Antibiotics were added at the following final concentration: kanamycin (Kn), 50 μg ml$^{-1}$ and ampicillin (Ap), 100 μg ml$^{-1}$.

### Genome sequencing and annotation

Single colonies were picked to inoculate in 5 ml LB medium. These cultures were grown for 18 h at 28°C and cells were then collected by centrifugation at 4000 rpm for 10 min. Genomic DNA was isolated from the cell pellets using the ChargeSwitch® gDNA Mini Bacteria Kit (Life Technologies) according to the manufacturer's instructions. Purified genomic DNA was quantified by TBS-380 fluorometer (Turner BioSystems Inc., Sunnyvale, CA). High quality DNA ($OD_{260/280}$ = 1.8~2.0, >10 μg) was used to do further research.

The genome sequence of *S. marcescens* 1912768R was determined by Shanghai Majorbio Bio-Pharm Technology Co., Ltd. (Shanghai, China) using the Pacbio sequencing system. For Pacific Biosciences sequencing, 8-10k insert whole genome shotgun libraries were generated and sequenced on a Pacific Biosciences RS instrument using standard methods. An aliquot of

**Table 1. Strains and plasmids used in this study.**

| Strains or plasmids | Characteristics | Reference or source |
|---|---|---|
| ***Escherichia coli*** | | |
| DH5α | *supE44 ΔlacU169(φ80 lacZΔM15) recA1 hsdR17 thi-1 relA1* | [34] |
| S17-1(λpir) | *recA pro hsdR* RP4-2-Tc::Mu-Km::Tn7 | [35] |
| ***Serratia marcescens*** | | |
| 1912768R | Isolated from rhizosphere soil in ginger fields | This study |
| 1912768WR | Mutant from 1912768R with reduced production of prodigiosin | This study |
| 1912768W | Mutant from 1912768R which is totally defective in the synthesis of prodigiosin | This study |
| Δ*rpoS* | Strain with the deletion of *rpoS* in 1912768R | This study |
| **Plasmids** | | |
| pMD19-T | Cloning and sequencing vector, Ap$^r$ | Takara |
| pK18mobsacB | Allelic replacement vector, Kn$^r$ | [36] |
| pBBR1MCS-2 | Broad-host-range cloning vector, Kn$^r$ | [37] |

8 μg DNA was spun in a Covaris g-TUBE (Covaris, MA) at 6,000 rpm for 60 seconds using an Eppendorf 5424 centrifuge (Eppendorf, NY). DNA fragments were then purified, end-repaired and ligated with SMRTbell sequencing adapters following manufacturer's recommendations (Pacific Biosciences, CA). Resulting sequencing libraries were purified three times using 0.45 x volumes of Agencourt AMPure XP beads (Beckman Coulter Genomics, MA) following the manufacturer's recommendations. The complete genome of *Serratia marcescens* 1912768R was deposited in GenBank under the accession number CP040350.

The genome sequence of *S. marcescens* 1912768WR and 1912768W was determined using the HiSeq2000 sequencing platform (Illumina Inc., San Diego, CA). For Illumina sequencing, at least 5 μg genomic DNA was used for each strain in sequencing library construction. Paired-end libraries with the insert size of ~300 bp were constructed according to the manufacturer's instructions (Bio-scientific, AIR™ Paired-End DNA Sequencing Kit). Subsequently, we sequenced 100 bp at each end using Illumina Hiseq 2000. The genomes of *Serratia marcescens* 1912768WR and 1912768W were deposited in GenBank under the accession number VBRI00000000 and VBRJ00000000, respectively.

The assembly was produced using SOAPdenovo software. The last circular step was checked and finished manually. The final assembly of 1912768R generated a circular genome sequence with no gap existed. Identification of predicted open read frames (ORFs) was performed using Glimmer version 3.02 (http://cbcb.umd.edu/software/glimmer/). ORFs with less than 300 base pairs were discarded. Then remained ORFs were queried against non-redundant (NR in NCBI) database, SwissProt (http://uniprot.org), KEGG (http://www.genome.jp/kegg/), and COG (http://www.ncbi.nlm.nih.gov/COG) to do functional annotation. In addition, tRNA were identified using the tRNAscan-SE (v1.23, http://lowelab.ucsc.edu/tRNAscan-SE) and rRNA were determined using the RNAmmer (v1.2, http://www.cbs.dtu.dk/services/RNAmmer/).

## Genetic manipulations and plasmid construction

All primers used were designed based on the genome sequence of *S. marcescens* 1912768R. The in-frame deletion of *rpoS* was constructed by two-step allelic replacement [38]. Briefly, two fragments flanking the *rpoS* gene were amplified by PCR. A DNA fragment containing 125 bp of the 5′ end of *rpoS* and 367 bp upstream of the ATG initiation codon was amplified from chromosomal DNA by PCR using primers F-rpoS-U1 and R-rpoS-D1. A DNA fragment containing 480 bp of the 3′ end of *rpoS* was amplified using primers F-rpoS-U2 and R-rpoS-D2. Both fragments were purified and fused in a subsequent PCR using primers F-rpoS-U1 and R-rpoS-D2. The fused segment was sequenced and ligated into the suicide vector pK18mobsacB using T4 DNA ligase (Takara, China). The resulting plasmid, pK18mobsacB-UD, was transferred by S17-1(λpir) into *S. marcescens* 1912768R by conjugation, and the transconjugants with the plasmid integrated into the chromosome by homologous recombination were selected on LB agar medium containing ampicillin and kanamycin at 28˚C. To create the *rpoS* deletion mutant strain, a second recombination event was counter-selected on LB agar containing 10% sucrose. Two external primers, F-rpoS-W and R-rpoS-W, anchored to the upstream and downstream regions of the *rpoS* gene, respectively, were adopted for the detection of deletion mutants by PCR. The deletion mutant was further confirmed by subsequent sequencing using the same primer pair. This deletion was in frame, and A126-G421 were removed from the 999 nucleic acid bases of the *rpoS* gene.

To complement the *rpoS* mutants, the *rpoS* gene of 1912768R with its native promotor (300 bp upstream of the *rpoS* ATG initiation codon) was amplified with primers F-rpoS-c and R-rpoS-c by PCR using the genomic DNA of *S. marcescens* 1912768R as the template. The PCR

product was then cloned into pBBR1-MCS2 to generate pBBR1-rpoS-R. The *rpoS* genes of *Serratia* sp. YD25, *S. plymuthica* AS9, *Serratia* sp. ATCC 39006 and *E. coli* K-12 *substr*. MG1655 together with the native promotor of 1912768R were synthesized and cloned into pBBR1-MCS2. The resulting plasmids were transferred by S17-1(λpir) into Δ*rpoS* 1912768R by conjugation to construct the *rpoS* complemented strain. Oligonucleotides used as primers for the *rpoS* deletion and complementation strains are listed in S1 Table.

## Sequence alignment

A homologous sequence database search was executed using BLASTp (https://blast.ncbi.nlm.nih.gov). Multiple sequence alignment was performed with ClustalX2.0 and visualized using ESPript 3.0 (http://espript.ibcp.fr/).

## Detection of extracellular enzymes, hemolysin and biofilm formation

Phenotypic assays were performed as described previously [39–42]. In all phenotypic plate assays, overnight cultures of the test strains were adjusted to an $OD_{600}$ of 1.0; 5 μl was spotted on the appropriate agar plates and incubated at 28°C. Prodigiosin was detected according to the method of Slater et al. [43]. Proteolytic activities were assayed on skimmed milk agar plates after 48 h as described [44]. Colloidal chitin was prepared per the method reported by Berger and Reynolds [45]; chitinolytic activities were assayed on 2% colloidal chitin plates after 48 h. Detection of hemolysin was indicated by the transparent zone on LB agar plates with 1% sheep erythrocytes (purchased from Beijing Land Bridge Co., Ltd.) for 72 h. Biofilm formation on 96-well plates was determined as previously described [46, 47] and quantified by measuring the absorbance at 575 nm using a plate reader (BioTek Synergy 2) after 72 h in LB medium.

## Motility assays

Swimming and swarming ability were assayed visually on LBA containing 0.35% (w/v) or 0.75% (w/v) agar, respectively. To assess swarming motility, 3 μl of an overnight culture in LB liquid medium was spotted on plates, allowed to dry and incubated for 16 h at 28°C. Transmission electron microscopy (TEM) was used to observe the cell shapes of strains 1912768R and Δ*rpoS*. Bacterial cells were cultured overnight in LB agar plates for 16 h. Colonies were washed in phosphate buffered saline (PBS) and placed on a carbon film-coated copper grid (230 mesh; Beijing Zhongjing Science and Technology Co., Ltd., Beijing, China). Finally, the bacterial liquid on the film was dried at 25°C and observed by TEM [47, 48]. At least three independent preparations were made for each group.

## Stress and competition assays

Bacteria grown overnight in LB medium were adjusted to an $OD_{600}$ of 1.0 with PBS buffer and treated as follows. For acid stress, 1 ml of an overnight culture was resuspended in 1 ml of 0.9% NaCl containing 175 mM acetic acid. Aliquots were removed every 5 to 20 min, diluted and plated on LBA. Oxidative stress was induced by treating a bacterial suspension with 0.9% NaCl containing 120 mM $H_2O_2$ for 5, 15, 30 and 45 min and subsequently diluting, and the bacterial culture was plated. Bacteria were subjected to osmotic stress by incubating for 0, 2, 4, 6 and 8 h in a 3 M NaCl solution. For heat stress, bacteria were treated at 50°C for 0, 5, 10, 15 and 20 min and plated. All plates were incubated for 24 h at 28°C followed by CFU counting. The results are shown as the percentage of the number of CFU/ml, with the CFU at time 0 being 100%.

Competition assays between the strains 1912768R and Δ*rpoS* mutants were performed under continuous culture in LB medium and M9 medium supplemented with 0.2% glucose at 28˚C for 48 h. The competition was started by mixing equal concentrations of each strain (at an $OD_{600}$ = 0.1). Samples were taken at time zero, 24 h and 48 h, and plated on LBA plates. CFU counts of each strain were determined following overnight incubation at 28˚C.

### RNA purification and qRT-PCR

Cell cultures (2 ml) were collected and centrifuged after16 h in LB medium and washed with RNase free water. RNA was extracted with the RNA Pure Bacteria Kit (with DNase I) (CWbio. Co., Ltd., Beijing, China). DNase treatment was performed with Cwbio DNase I at 28˚C for 15 min prior to the concentration step. RNA was quantified using a Nanodrop ND1000 (Nano-Drop Technologies), and the quality was evaluated using an Agilent Bioanalyzer (Agilent Technologies). First-strand cDNA was synthesized using the PrimeScript RT Reagent Kit with gDNA Eraser (TaKaRa, Shiga, Japan) according to the manufacturer's instructions [15]. Each qRT-PCR was conducted in a final volume of 25.0 μL that contained 8.5 μL of sterile water, 1 μL of the corresponding primers (10 μM), 2.0 μL of suitable cDNA as the template, and 12.5 μL of SYBR Premix ExTaq II (TaKaRa, Shiga, Japan). All reactions were run for 40 cycles, of 3 s at 95˚C and 30 s at 60˚C. The melting curve program was set for 5 s at 95˚C, 1 min at 60˚C, and 15 s at 95˚C. Gene expression levels were measured using the 16S gene as a control and normalized to the parental strain 1912768R. Oligonucleotides used as primers for quantitative PCR are listed in S1 Table.

### Statistical analysis

Experiments were performed at least twice with a minimum of three biological replicates. GraphPad Prism software was used to perform Student's t-tests and one-way ANOVA with Tukey's post-test. Significance was set at $p < 0.05$.

## Results

### Selective advantage for nonpigmented mutants during prolonged stationary phase

To verify whether prolonged incubation would enrich the *de novo* emergence of prodigiosin-defective individuals, we performed a series of long-term incubations of the WT strain 1912768R in lysogeny broth (LB) medium. As shown in Fig 1A, cells incubated in batch culture began to lose viability at day 2, marking the transition from the stationary phase to the death phase. More than 99% of the cells died, and the number of viable cells became stable on day 16. Prodigiosin-deficient mutants were first isolated on day 6 and enriched with the extension of culture time. The arrow indicates the time after which cells expressing the GASP phenotype were observed, and the number of GASP mutants (white strains) reached a maximum around day 16. This result suggested that the loss of prodigiosin production might contribute to the survival of white mutants during prolonged stationary phase. To verify whether the selective advantage for nonpigmented mutants is incidental, 10 single colonies of *S. marcescens* 1912768R were grown for 16 days. As shown in Fig 1B, all nonpigmented mutants were observed around day 6, and the white strains were enriched during prolonged incubation.

To explore the underlying mechanisms of loss of prodigiosin production, a white colony of *S. marcescens* 1912768R was randomly selected after 16 days of prolonged incubation of under laboratory conditions. A pigment-defective mutant (pink) was also isolated at the same time. These two mutants were named 1912768WR (pink) and 1912768W (white) (Fig 1C).

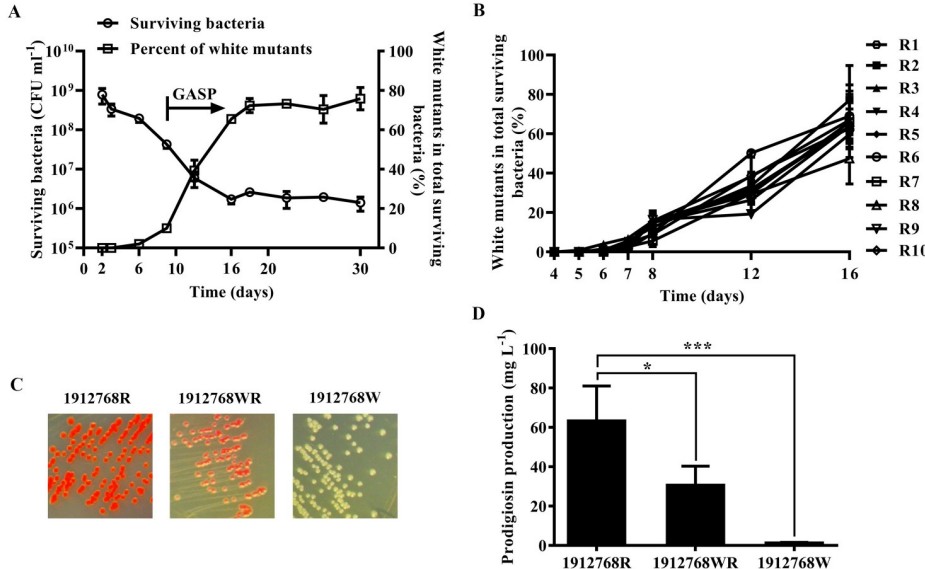

**Fig 1. Prolonged incubation of *S. marcescens* 1912768R and isolation of mutants with prodigiosin biosynthesis defects.** (A) Changes in the number of all viable strains and percentage of emerged white strains during the prolonged incubation of WT. (B) Frequency of *de novo* emergence of prodigiosin-defective mutants. R1-10 represent ten single colonies of WT 1912768R cultured independently under the same culture conditions. (C) Representative images of the WT strain 1912768R and the two mutants. (D) Quantification of prodigiosin production in LB medium. *, P < 0.05; ***, P < 0.001.

Prodigiosin was extracted and measured from cultures of the WT and two mutant strains. The prodigiosin levels were significantly lower in the mutants as shown in Fig 1D.

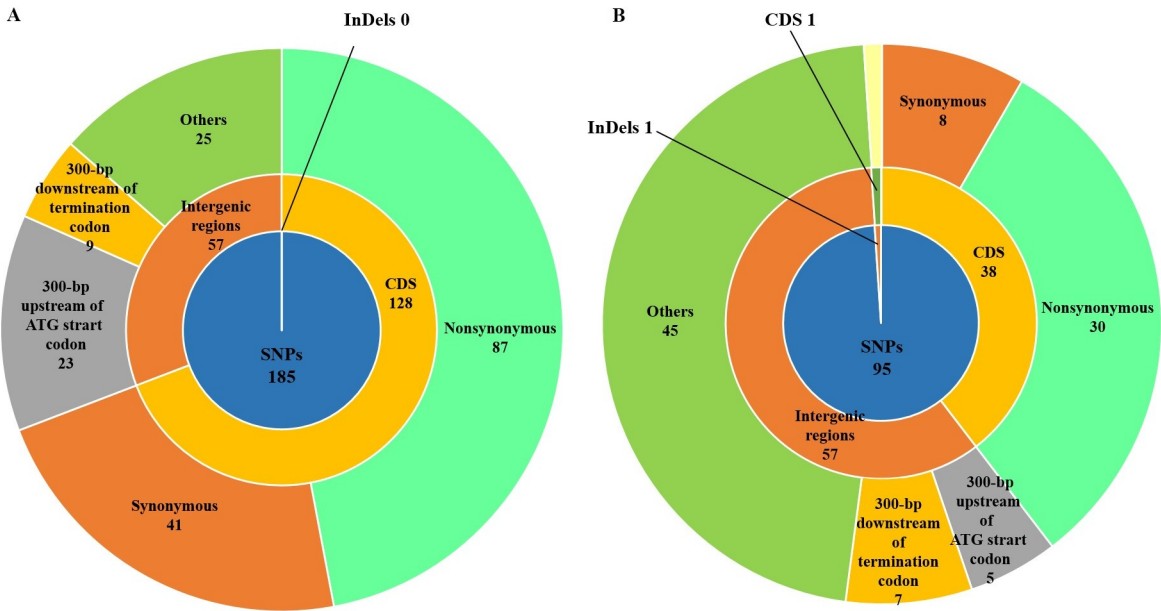

**Fig 2. Distribution of SNPs and InDels in *S. marcescens* strain 1912768WR and 1912768W compared to WT.** CDS refers to coding DNA sequence. Each sector stands for the number and location of SNPs and Indels between the corresponding mutant and the WT strain.

## RpoS was screened as a candidate regulator of prodigiosin production

Comparative genomic analysis was performed to further explain the underlying molecular mechanism of the deficiency in prodigiosin production. *De novo* sequencing of 1912768R was performed and the complete genome of 1912768R (GenBank accession number: CP040350) consists of a 5,117,289 bp (GC content 59.75%) single circular chromosome (S1 Fig). A total of 4927 protein-encoding genes were predicted and annotated in four databases namely the Kyoto Encyclopedia of Genes and Genomes (KEGG), the National Center for Biotechnology Information non-redundant (NR), UniProt, and Gene Ontology (GO) databases. The genomes of 1912768WR (GenBank accession number: VBRI00000000) and 1912768W (GenBank accession number: VBRJ00000000) were sequenced. The general features of the genomes are presented in S2 Table.

Comparative analysis revealed 185 and 95 nonsynonymous SNPs in 1912768WR and 1912768W, respectively, and one deletion in 1912768W (Fig 2). Of the 185 SNPs in 1912768WR, 128 were located in coding DNA sequences (CDSs) and 57 were in intergenic regions in the promoters and terminators around protein-coding genes which could influence the transcriptional level of the corresponding genes [49]; of the 95 SNPs in 1912768W, 38 and 57 were located in CDSs and intergenic regions, respectively. The only deletion was in the *rpoS* gene of 1912768W, resulting in a frameshift mutation.

BlastP analysis revealed that the *pig* cluster from 1912768R was a type I *pig* cluster (S2 Fig). Since none of the SNPs/InDels were mapped in the *pig* cluster, this meant that mutations in transcriptional regulators that regulate prodigiosin production might lead to the observed pigmentation. The number of nonsynonymous mutations that were located in transcriptional regulators were 8 and 3 for strains 1912768WR and 1912768W, respectively. In addition, one nonsynonymous SNP was found in *rpoS* at site 62 (T/A→C/G) in 1912768WR, and a deletion was found in *rpoS* at sites 185–186 (CG, 2-nucleotide deletion) in 1912768W. BlastP analysis of RpoS from different *Serratia* sp. with different *pig* cluster types and *E coli* was performed. The sequences were highly conserved with identities of 99.70% (*Serratia* sp. YD25), 97.89% (*Serratia plymuthica* AS9), 91.87% (*Serratia* sp. ATCC 39006) and 94.58% (*Escherichia coli* K12). The mutations of RpoS within the two mutants were located at sites 62 and 185–186, and these mutations were highly conserved (Fig 3). RpoS was proven to be a negative regulator in prodigiosin regulation in strain *Serratia* sp. ATCC 39006 [17]; however, this result suggested that RpoS from *S. marcescens* 1912768R might be a positive regulator of prodigiosin production, and the mutation in *rpoS* might be causing the deficiency in prodigiosin production.

## RpoS positively regulates the production of prodigiosin in *S. marcescens* 1912768R

According to the analysis of the function of RpoS in prodigiosin production above, the regulatory role of RpoS in pigmentation was further characterized. A 296 bp segment (A126-G421) was deleted from the 999 bp *rpoS* gene of 1912768R. Prodigiosin of the WT strain and Δ*rpoS* mutant strains with the complementation plasmid pBBR1-*rpoS*-R or vector negative control was extracted and measured after two days of fermentation. The prodigiosin levels were significantly lower in the Δ*rpoS* mutant and could be complemented in *trans* (Fig 4A). This effect could stem from reduced transcriptional expression of the prodigiosin biosynthesis operon. Quantitative reverse transcriptase PCR (qRT-PCR) analysis was used to evaluate the relative transcription of the *pig* cluster in the WT and Δ*rpoS* mutant strains. More than a 20-fold and 6-fold reduction in the *pigA* transcript level was observed in the Δ*rpoS* mutant, suggesting that the pigment defect of the Δ*rpoS* mutant is largely due to the decreased transcriptional level of the *pig* cluster (Fig 4B). In *Serratia* sp. ATCC 39006, the *rpoS* mutant was unable to grow in

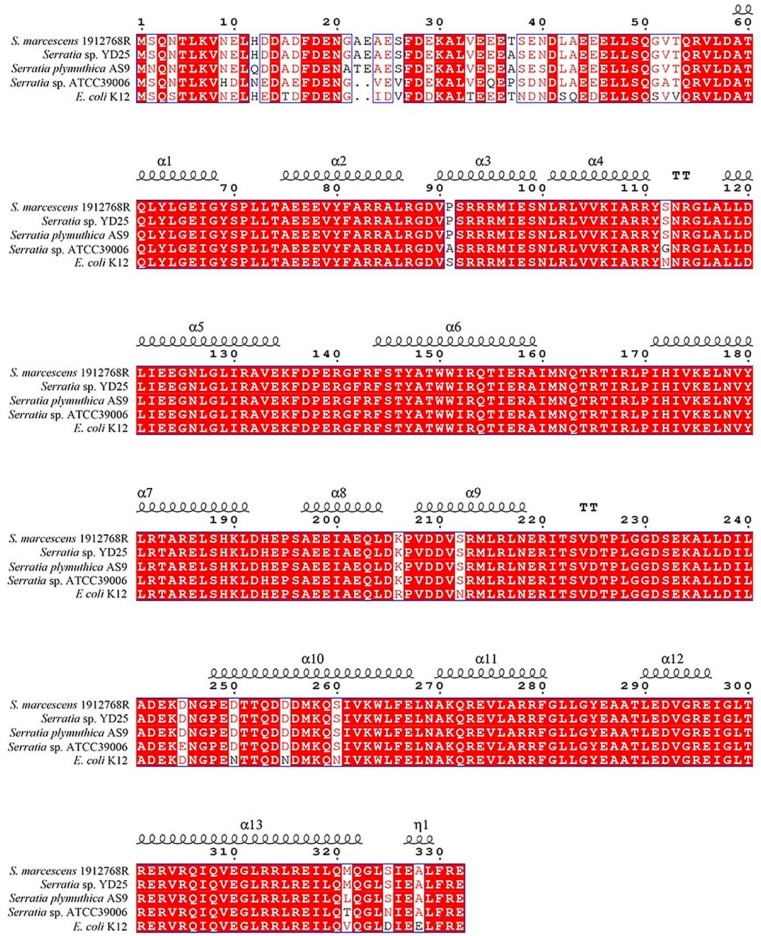

**Fig 3. Sequence alignment of RpoS.** Multiple sequence alignment of RpoS from *S. marcescens* 1912768R, *Serratia* sp. YD25, *S. plymuthica* AS9, *E. coli str.* K-12 substr. MG1655 and *Serratia* sp. ATCC 39006. The alpha helix, random coil and beta turn are represented as α, η and T, respectively. Conserved sequences are indicated with a red box, and similar sequences are indicated using a colored background.

minimal medium (MM) with glucose as the sole carbon source, suggesting that the mutant was auxotrophic for essential amino acids [17]. Growth analysis of the *rpoS* mutant of *S. marcescens* 1912768R in LB and MM with glucose as the sole carbon source showed no difference, indicating that prodigiosin differences were not due to growth and that the role of RpoS in growth is different (Fig 4C and 4D).

To further confirm the regulatory effect of RpoS on the production of prodigiosin, *rpoS* genes from strains with different *pig* cluster types and *E. coli* K12 together with the promoter from 1912768R were synthesized. The synthesized segments were introduced into the Δ*rpoS* mutant strain on a multicopy plasmid. With the expression of *rpoS*, the pigment and colony morphology phenotypes of the Δ*rpoS* mutant were restored to those of the WT strain to some extent (Fig 4E). These results support the idea that RpoS is a positive regulator of prodigiosin production in *S. marcescens* 1912768R.

## The disruption of *rpoS* has pleiotropic effects on *S. marcescens* physiology

The prodigiosin production regulator, RbsR in S39006, is a pleiotropic regulator of motility, virulence, siderophore, exoenzymes and antibiotic production [50]. MetR controls prodigiosin

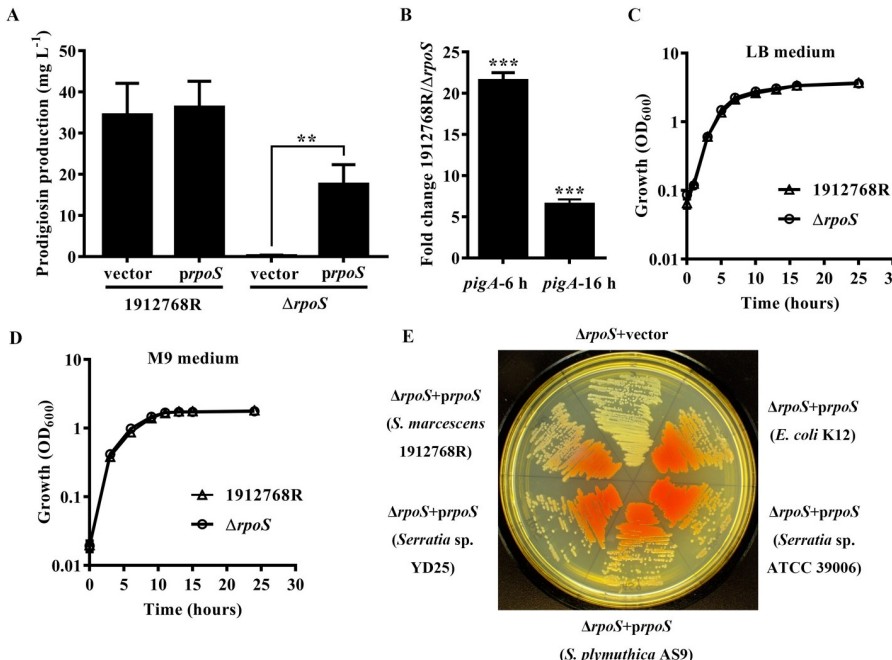

**Fig 4. RpoS is a positive regulator of prodigiosin production, and the function of RpoS is conserved in *S. marcescens* 1912768R.** (A) The *rpoS* mutant colony and color phenotypes were complemented by *rpoS* from *S. marcescens* 1912768R. (B) qRT-PCR analysis revealed reduced expression from the prodigiosin biosynthetic locus. (C-D) Growth curves of the WT strain 1912768R and the Δ*rpoS* mutant in LB or M9 minimal medium supplemented with 0.2% glucose. (E) Complementation of Δ*rpoS* mutant defects by *rpoS* genes from *Serratia* sp. strains with different *pig* cluster types and *E. coli*.

production, methionine biosynthesis, cell motility, $H_2O_2$ tolerance, heat tolerance, and exopolysaccharide synthesis in *Serratia marcescens* JNB5-1 [48]. It seems that the regulator of prodigiosin production has pleiotropic effects on strain physiology in general. As the *rpoS* mutation regulated prodigiosin production in *S. marcescens* 1912768R, we postulated that it might also regulate other phenotypes and modes of motility. In addition to the abolishment of prodigiosin production, biofilm formation, hemolysis and proteinase secretion also decreased compared to WT strain 1912768R, and the secretion of lipase, chitinase and siderophores increased (Fig 5A). These observations suggested that prodigiosin production might have a correlation with the different phenotypes and further highlighted the extent of the pleiotropy caused by the *rpoS* mutation in 1912768R.

Tests were performed to determine whether RpoS played a role in the regulation of motility. As shown in Fig 5B, the swimming and swarming ability of the Δ*rpoS* mutant increased significantly. TEM analysis of the Δ*rpoS* mutant cells after 12 h of plate culture also demonstrated an increase in flagella and elongated shape, which contributed to the increased mobility. To further identify the role of RpoS in the regulation of motility, qRT-PCR was performed. The transcript levels of the *flhC* and *flhD* genes in the Δ*rpoS* mutant strain increased by several times compared to those in the WT strain (Fig 5C). This increase in expression of the *flhDC* operon correlated with an increase in the expression of the flagellar gene *fliC* based on qRT-PCR; this result is consistent with the increase in mobility of the Δ*rpoS* mutant.

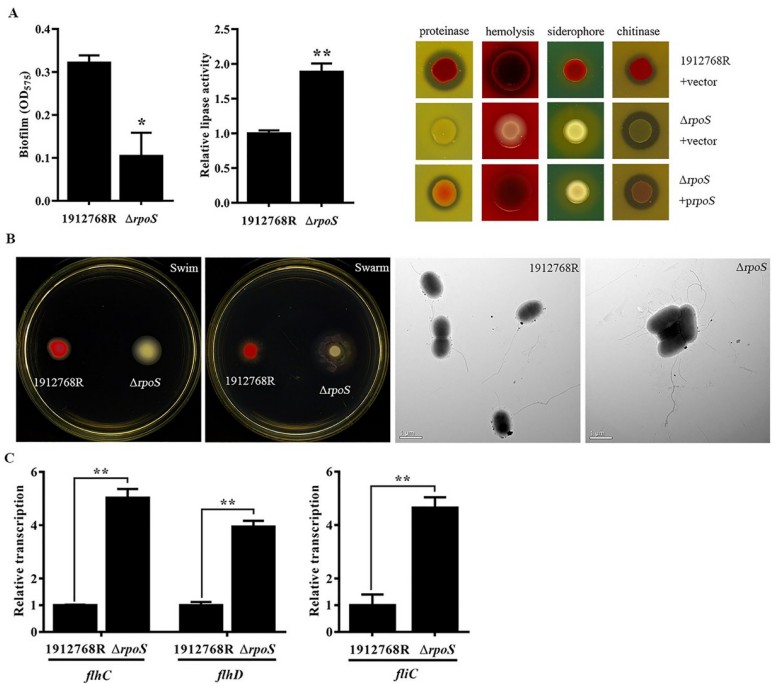

**Fig 5. The disruption of *rpoS* has diverse effects on the physiology and motility of 1912768R.** (A) Diverse physiology of biofilm formation, lipase and proteinase secretion, hemolysis ability, production of siderophores and chitinase activeity. (B) Mobility and TEM analysis of the WT strain and the *rpoS* mutant. (C) Changes in the transcript levels of motility-related genes. qRT-PCR of *fliC* and *flhCD* gene expression, n = 4. Mean and SD are shown. *, P < 0.05; **, P< 0.01.

## The Δ*rpoS* mutant displayed impaired resistance to stresses but higher competence than its isogenic WT strain

The physiological role of bacterial prodigiosin in vivo remains undefined. However, a number of reports have suggested potential functions for prodigiosin that may provide advantages such as antagonistic activity to other strains (biotic stress) [51] and continuously challenging natural environments (abiotic stress) [52]. In *E. coli* and *Salmonella enterica* serovar Typhi, *rpoS* regulates resistance to stresses including high osmolarity, oxygen free radicals, acid and other biotic stresses [20–22]. The attenuation function of *rpoS* or even *rpoS* mutants confers a reduction in stress resistant but promots of nutritional competence because bacteria allocate limited resources to growth rather than self-protection when suffering starvation and vice versa [23]; therefore, there is a SPANC trade-off in *E. coli*, which is an adaptive behavior during prolonged starvation [53].

The role of RpoS in stress resistance was explored. The survival rates of 1912768R and the pigmentless Δ*rpoS* mutant were characterized after exposure to diverse stress conditions, including 10 mM $H_2O_2$, 50°C, 175 mM acetic acid and 20% NaCl. Plate counts at different time intervals after exposure to the corresponding stress were deterimined to evaluate the viabilities of the WT 1912768R and Δ*rpoS* mutant strains. The results showed that the survival rates of 1912768R and the Δ*rpoS* mutant decreased with increasing environmental pressure; in addition, the Δ*rpoS* mutant was more sensitive to oxidative, osmotic, heat and acid stress (Fig 6A–6D). These results proved the positive regulatory role of RpoS in stress resistance and indirectly suggested that prodigiosin might provide an advantage for *S. marcescens* in stress resistance.

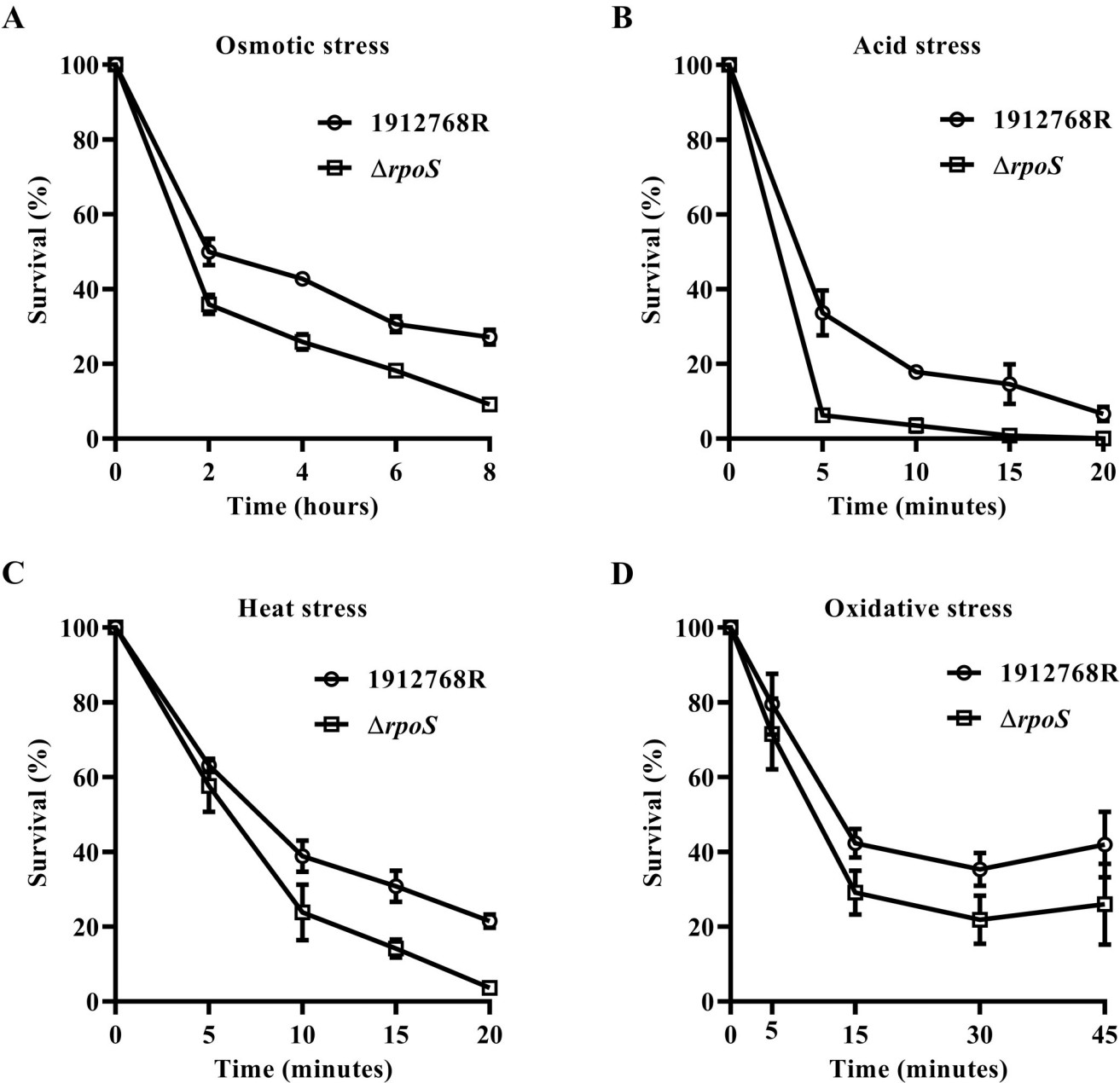

**Fig 6. Stress resistance assays of *S.marcescens* 1912768R and the Δ*rpoS* mutant.** The WT strain 1912768R and in-frame deletion Δ*rpoS* mutant were exposed to 3 M NaCl solution (A), 175 mM acetic acid (B), 50˚C (C) and 120 mM $H_2O_2$ (D). The survival percentage was obtained by dividing the surviving population by the initial population, which corresponded to 100%. Data are mean and SD of three independent experiments.

An indirect but frequently used test for outer membrane permeability is susceptibility to SDS and Cm, as outer membrane exclusion is an important component of protection against these compounds [54–57]. The survival rate of the Δ*rpoS* mutant decreased compared to that of the WT strain when treated with SDS and Cm, indicating that the permeability of the membrane of the Δ*rpoS* mutant increased (Fig 7A and 7B), which would promote nutritional competence. The general membrane permeability of *E. coli* is determined by the type of major porin proteins, OmpC and OmpF, which act as molecular sieves; OmpC channels are more

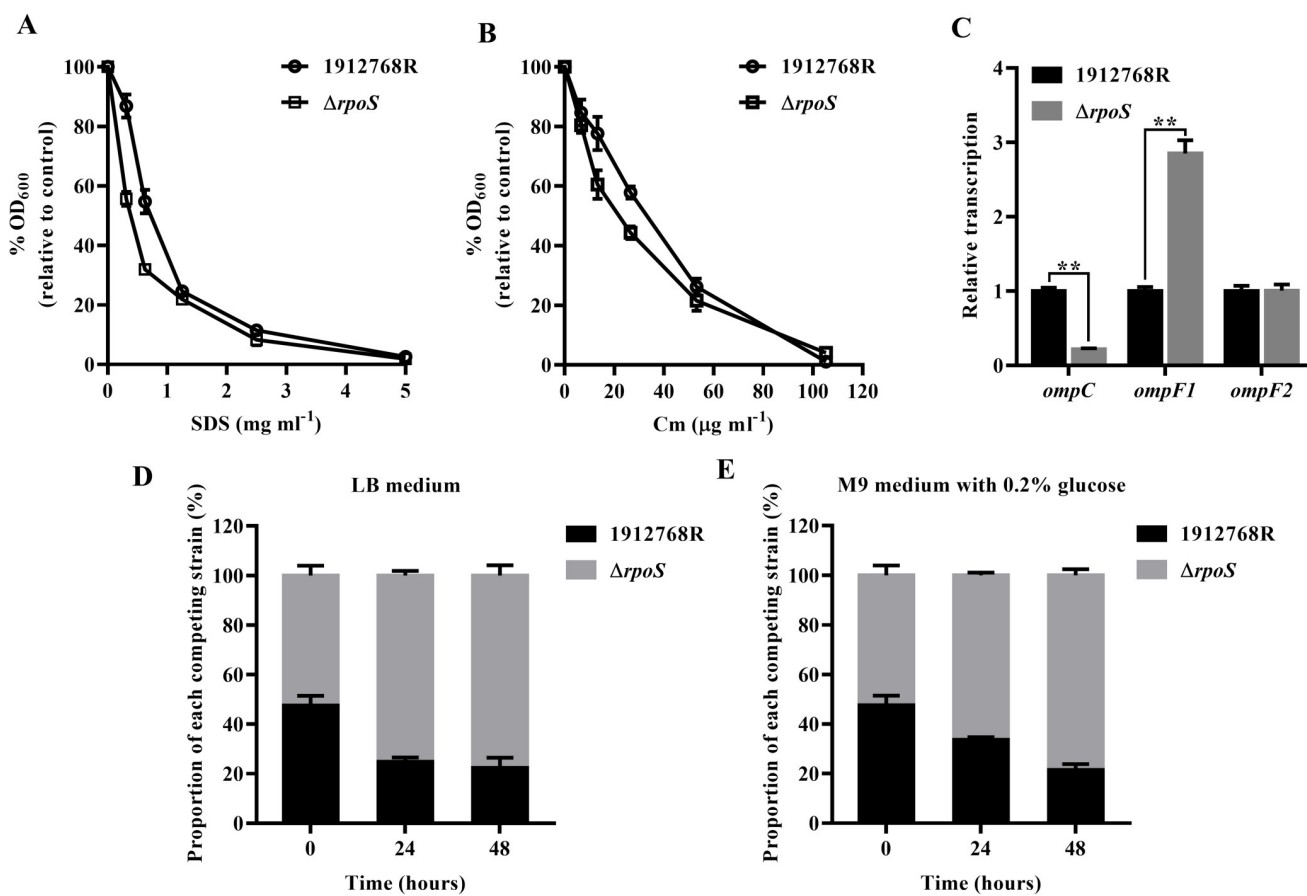

**Fig 7. Effect of *rpoS* on membrane permeability and relative fitness of *S. marcescens* 1912768R and the Δ*rpoS* mutant.** (A-B) Growth of the 1912768R and Δ*rpoS* mutant strains on 96-well plates containing SDS or Cm. (C) Change in transcript levels of *luxS* and porin-related genes (n = 4). (D-E) Effect of *rpoS* on the fitness of *S. marcescens*. *S. marcescens* 1912768R with theΔ*rpoS* mutant at a 1:1 ratio and grown for 24 and 48 h in LB or M9 medium. Samples were removed, diluted and plated on LBA plates for CFU determination. The number of 1912768R and Δ*rpoS* mutant colonies was verified by the different color of the colonies. Mean and SD are shown. *, P < 0.05; **, P < 0.01.

restrictive than OmpF channels [57, 58]. Bacteria containing only OmpC are more resistant to toxic hydrophilic agents than those containing only OmpF, while the transcription of *ompF* increases under glucose limitation, improving the permeability of substrates at low concentrations [54]. To further investigate membrane permeability, the transcript level of *ompF/ompC* was identified. Fig 7C shows that the transcript level of *ompF* increased in the Δ*rpoS* mutant strain, while that of *ompC* decreased; which is consistent with the increase in cell membrane permeability. These results suggest that RpoS positively regulates membrane permeability. The higher the cell membrane permeability of the strain is, the more sensitive the strain is to pressure, and the easier it is for nutrients to pass through.

To evaluate the contribution of *rpoS* to the fitness of WT 1912768R and the Δ*rpoS* mutant, competition assays were performed in LB and M9 minimum medium with supplemented with 0.2% glucose. Equal numbers of bacteria in the exponential phase were mixed and incubated with the corresponding medium. By coculturing mixtures of 1912768R and the Δ*rpoS* mutant and plate counting based on the different color of the colonies on LB agar plate (the nonpigmented strain mutated from 1912768R appeared on the 6[th] day), we found that the Δ*rpoS* mutant performed better than the WT strain in both LB and M9 minimum medium at 24 h

and 48 h, as shown in Fig 7D and 7E. Hence, *rpoS* did have a clear deleterious effect on the fitness of *S. marcescens* under laboratory conditions.

## Discussion

In this study, we identified that RpoS was a positive regulator of prodigiosin biosynthesis in *S. marcescens* 1912768R; mutation of *rpoS* introduced pleiotropic phenotypes. We also made the case that mutation of *rpoS*, which led to the loss of prodigiosin production during prolonged stationary phase, was a trade-off between self-preservation and nutritional competence in *S. marcescens* 1912768R.

The genetic results presented here indicate that the RpoS protein from several species of *Serratia* and *E. coli* is highly structurally conserved. Therefore, it was somewhat surprising that the function of RpoS in the regulation of prodigiosin production is different between *S. marcescens* 1912768R and *Serratia sp*. ATCC 39006. Random transposon insertion of a 533 bp fragment after the start codon of *rpoS* in *Serratia* sp. ATCC 39006 resulted in increased prodigiosin production [17]. In this paper, deletion of the 296 bp fragment revealed that RpoS from *S. marcescens* 1912768R was a positive regulator of prodigiosin biosynthesis. Complementation of RpoS from different strains with the native promoter of *rpoS* from strain 1912768R restored the pigmentation, and it seems that the promoters of the *pig* cluster are more important than the genes for the regulation of prodigiosin production. Currently, the regulatory pathway of RpoS in the biosynthesis of prodigiosin is unknown. Therefore, some theoretical basis is still needed to fully clarify the reason why the regulatory roles of RpoS in regulating production of prodigiosin are different in *Serratia marcescens* 1912768R and strain *Serratia* sp. ATCC 39006.

One reason for the opposite regulatory roles of RpoS may be due to the diverse types of *pig* clusters. BlastP analysis showed that the *pig* cluster of 1912768R was a type I *pig* cluster, which is the main form of the prodigiosin coding cluster existing in the majority of strains and is different from the unique *pig* cluster that only exists in *Serratia* sp. ATCC 39006 (S2 Fig). Although the global arrangement of the *pig* clusters in 1912768R and *Serratia* 39006 is very similar, the intergenic region upstream of *pigA* is markedly different. Transposon insertion mutations in the region 200–320 bp upstream of *pigA* in *Serratia* 39006 cause hyperpigmentation [59]. The promoter of the *pig* cluster has been mapped, and a different promotor has been proposed in the 488 bp region upstream of *pigA* in *S. marcescens* Sma 274 [60]. In *S. marcescens* 1912768R, this region is 407 bp compared to 924 bp in *Serratia* 39006, indicating that promoter regions in *pig* clusters might be different.

Another conspicuous difference between 1912768R and ATCC 39006 is the quorum sensing (QS) system. In *Serratia* 39006, transcription of the *pig* cluster is regulated by QS via the *smaI/R* genes and the N-acyl homoserine lactones (AHLs) N-butanoyl-L-homoserine lactone (BHL) and N-(hexanoyl)-L-homoserine lactone (HHL) [61], while *smaI/R* homologues do not exist in strain 1912768R. The enzyme LuxS is responsible for the production of autoinducer-2 (AI-2), a molecule that has been implicated in QS in many bacterial species. Mutation of *luxS* from *Serratia* sp. ATCC 39006 had no influence on prodigiosin biosynthesis while the *luxS* mutant of *S. marcescens* ATCC 274 showed decreased prodigiosin production [18]; in addition, the decreased transcriptional level of *rpoS* showed no influence on the transcription of *luxS* in *Serratia* sp. ATCC 39006, suggesting that the regulation of RpoS in prodigiosin production is independent of LuxS [16]. In this study, the transcriptional level of *luxS* decreased significantly in the Δ*rpoS* mutant compared to the WT strain 1912768R (S3 Fig), indicating that the regulation of prodigiosin production in *S. marcescens* 1912768R by RpoS might be through LuxS (Fig 8), and this mechanism of regulation is unknown in *Serratia* sp. ATCC 39006 [17].

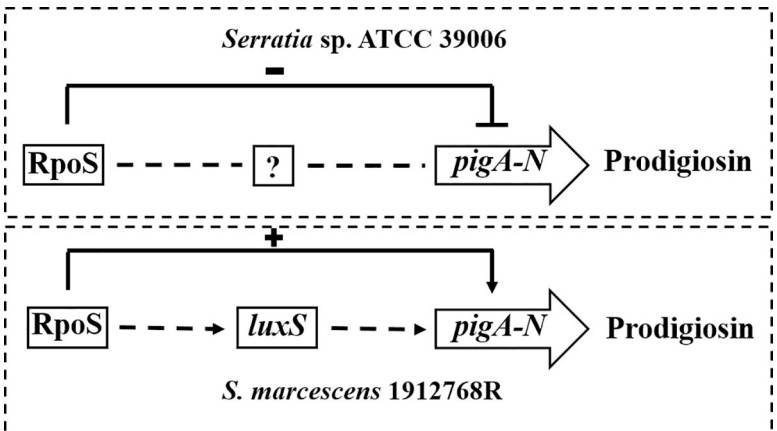

**Fig 8. Proposed model of prodigiosin production regulation in *Serratia* sp. ATCC 39006 and *S. marcescens* 1912768R by *luxS* and *rpoS*.** Positive regulation is depicted by pointed arrowheads. Dashed lines indicate putative regulation.

Finally, the different roles of RpoS may be due to the diverse function of RpoS in the regulation of secondary metabolites. In *Pseudomonas* species, the regulatory effect of RpoS on secondary metabolites is diverse or even opposite in some species [62], illustrating that the diversity of the regulatory effect of RpoS is not unique. As the regulatory pathway of RpoS in the biosynthesis of prodigiosin is unknown and complicated, further research is still needed to illuminate the reasons for the distinct regulatory roles of RpoS in the two strains.

The functional diversity of RpoS in *Serratia* species can also be observed in the regulation of other phenotypes. The *Serratia* sp. ATCC 39006 Δ*rpoS* mutant could not be cultured in minimal medium; on the contrary, the growth rate of the Δ*rpoS* mutant did not change compared to that of the WT strain 1912768R, which might be due to the functional diversity of RpoS in *Serratia* species. The Δ*rpoS* mutant showed increased swimming motility, which is different from the unchanged swimming motility observed in the *rpoS* mutant of *Serratia* sp. ATCC 39006. Swarming is dependent on both biosurfactants, such as serratamolide, and flagellar motility [7]. Serratamolide is hemolytic, and we observed an obvious decrease in hemolysis of the Δ*rpoS* mutant. However, the amount of swarming increased, suggesting that the increase in *flhDC* expression increases motility to overcome the reduction in serratamolide. The TEM results also revealed that Δ*rpoS* cells have a swarmer morphology characterized by an elongated shape and additional flagella that could account for the increased zone of swarming even with an apparent reduction in biosurfactant.

The stress resistance of the Δ*rpoS* mutant in our study decreased, which was similar to that of the Δ*rpoS* mutant of *E. coli* [20, 23]. We also observed an increase in membrane permeability as shown in Fig 7. In *E. coli*, the general membrane permeability is determined by the porin proteins OmpC/OmpF [63]; the difference between them is that OmpC channels are more restrictive, protecting cells against toxic hydrophilic agents, but OmpF channels offer higher open access to diffusible nutrients and are the most open porin in *E. coli* [64]. In our study, the transcript level of *ompF* increased in the Δ*rpoS* mutant strain, while that of *ompC* decreased. As no research has shown that OmpC and OmpF in *S. marcescen*s act similarly to *E. coli* porins, we speculate that the increase in cell membrane permeability is due to transcriptional changes in the corresponding porin genes. Reducd stress resistance and increased nutritional competence make the Δ*rpoS* mutant outcompete its parent strain, indicating an SPANC trade-off in *S. marcescens* 1912768R driven by the mutation of *rpoS*.

In summary, this study introduces the novel regulatory role of RpoS from *S. marcescens* on the secondary metabolite prodigiosin and enhances the knowledge on the regulation by RpoS. Additionally, this study confirms that *rpoS* mutants emerged during prolonged stationary phase are the major cause of the loss of prodigiosin production, and this appears to be an elaborate evolutionary strategy driven by the trade-off in *S. marcescens* 1912768R.

## Supporting information

**S1 Table. Primers used in this study.**
(PDF)

**S2 Table. General genome features of *S. marcescens* strains 1912768R, 1912768WR and 1912768W.**
(PDF)

**S1 Fig. Circular diagrams of *S. marcescens* 1912768R genome.** The outermost circle is the genome size, each scale is 0.1 Mb; the second and third circles are the CDS on the positive and negative strands, and different colors indicate the different COG functional classification of the CDS; the fourth circle is rRNA and tRNA; the fifth circle is the GC content. The outward red part indicates that the GC content in this area is higher than the average GC content of the whole genome. The higher the peak, the greater the difference from the average GC content. The inward blue part indicates that the regional GC content is lower than the average GC content of the whole genome; the innermost circle is the GC-skew value.
(TIF)

**S2 Fig. Comparison of *pig* cluster from *S. marcescens* 1912768R and *Serratia* sp. ATCC 39006.**
(TIF)

**S3 Fig. Relative transcription of gene *rpoS*.** Relative transcription of gene *rpoS* from WT strain, Δ*rpoS* and *rpoS* complemented strain at 6 h (a) and 16 h (b).
(TIF)

## Author Contributions

**Conceptualization:** Han Qin, Hui Xu, Yi Cao.

**Data curation:** Han Qin, Ying Liu, Jia Jiang, Hui Xu.

**Formal analysis:** Han Qin, Ying Liu, Hui Xu.

**Funding acquisition:** Dairong Qiao, Hui Xu.

**Investigation:** Han Qin, Ying Liu.

**Methodology:** Han Qin, Xiyue Cao, Hui Xu.

**Project administration:** Han Qin.

**Software:** Jia Jiang.

**Supervision:** Dairong Qiao, Hui Xu.

**Validation:** Ying Liu, Xiyue Cao, Dairong Qiao.

**Visualization:** Ying Liu, Xiyue Cao.

**Writing – original draft:** Han Qin.

**Writing – review & editing:** Han Qin, Weishao Lian, Dairong Qiao, Hui Xu, Yi Cao.

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
