## [Decision Letter · Decision Letter 0]

20 Feb 2020

PONE-D-20-03043

Regulation of RpoS in production of prodigiosin and trade-off during prolonged stationary phase in Serratia marcescens 1912768R

PLOS ONE

Dear Dr Yi Cao,

Thank you for submitting your manuscript to PLOS ONE. After careful consideration by two experts in the field, we feel that it has merit but does not fully meet PLOS ONE’s publication criteria as it currently stands. Therefore, we invite you to submit a revised version of the manuscript that addresses the various points raised during the review process.

We would appreciate receiving your revised manuscript by Apr 05 2020 11:59PM. To enhance the reproducibility of your results, we recommend that if applicable you deposit your laboratory protocols in protocols.io, where a protocol can be assigned its own identifier (DOI) such that it can be cited independently in the future. For instructions see: http://journals.plos.org/plosone/s/submission-guidelines#loc-laboratory-protocols

We look forward to receiving your revised manuscript.

Kind regards,

Marie-Joelle Virolle, PhD

Academic Editor

PLOS ONE

Journal Requirements:

5. Please amend your list of authors on the manuscript to ensure that each author is linked to an affiliation. Authors’ affiliations should reflect the institution where the work was done (if authors moved subsequently, you can also list the new affiliation stating “current affiliation:….” as necessary).

Reviewers' comments:

Reviewer's Responses to Questions

**Comments to the Author**

1. Is the manuscript technically sound, and do the data support the conclusions?

Reviewer #1: Yes

Reviewer #2: Partly

2. Has the statistical analysis been performed appropriately and rigorously? 

Reviewer #1: Yes

Reviewer #2: Yes

3. Have the authors made all data underlying the findings in their manuscript fully available?

Reviewer #1: Yes

Reviewer #2: Yes

4. Is the manuscript presented in an intelligible fashion and written in standard English?

Reviewer #1: No

Reviewer #2: No

5. Review Comments to the Author

Reviewer #1: Review of PONE-D-20-03043 for PLoS One, “Regulation of RpoS in production of prodigiosin an trade-off during prolonged stationary phase in Serratia marcescens 1912768R.

This study describes the enrichment of pigmentless mutants of S. marcescens during prolonged stationary phase. The basis of this loss of pigment was found to be mutation of the rpoS sigma factor gene, which is in opposition to what was previously shown by the Salmond group with another Serratia species. The defect in pigmentation was shown to be due to loss of transcription of the pig operon and not due to changes in growth. The rpoS mutant pigment defect was complemented by wild-type rpoS genes from a variety of Serratia species and Escherichia coli. Mutation in rpoS was demonstrated to have pleiotropic effects including increased susceptibility to a number of stresses. Additional data suggested that there is differential expression of OMPs and a selective growth advantage of the rpoS mutant in LB and minimal media.

On the positive side, the figures are well put together. The data for the most part is good, particularly exciting are figure 1A/B and figure 4. There is a lot of room for improvement, particularly with respect to the major issues in the writing and some concerns with interpretation.

Criticisms:

1. The major issue are the numerous writing issues (grammar and spelling) that weaken the manuscript. I have listed many below, but there are too many to list.

2. Line 21. Please change “which is” to “could be” – this is important because the results of this study suggest that loss of pigmentation will be a problem for industry, but this has not been reported as suggested by the current phrase.

3. Line 97. inconsistent with line 238. LB should be Lysogeny Broth, but Luria-Bertani is also accepted, but please be accepted.

4. Table 1. Inconsistent description of DH5a and S17-1 – please add the genotype for S17-1.

5. Table 1. Cut “widely used” for two plasmids, also replace knockout vector with “allelic replacement vector”

6. Line 148-151. Please be more specific about the mutant allele. For example, what base pairs are left.

7. Line 158-9. Replace “double-crossover recombination” with “second recombination”. Double cross-over is incorrect in this allelic replacement approach.

8. Methods line 183. Please describe what is meant by “treated chitin” and/or add a reference.

9. Line 185. What is meant by “skim sheep blood”?

10. Line 215-20. Please describe whether DNase was used in RNA preparation.

11. Line 234-5. Suggest changing the title to “Selective advantage for non-pigmented mutants during prolonged stationary phase”. Line 236. Change “cause” to “enrich for”

12. Figure 2. Should say “intergenic region” rather than “intergenic gene”

13. Line 289 – change “type to types”. Moreover, give some information on the amino acid identity of the RpoS protein.

14. Line 308 – consider adding the fold-change. It is hard to determine this looking at the linear scale of figure 4B.

15. Line 310. Consider adding “indicating that prodigiosin differences were not due to growth”

16. Line 320. It was very good using the same promoter for each rpoS gene. Please indicate that all genes were expressed from the rpoS promoter from 1912768R.

17. Line 339-340. “increased by several orders of magnitude” – this suggests that the Y-axis of Figure 5C was a log10 scale – please clarify whether it is log10 or linear.

18. Line 359-365. The loss of permeability using these assays is not well described. Are these well-established assays for permeability? I would suggest adding references for these.

19. Line 363-5. Is there any information on the predicted pore size of OmpC and OmpF?

20. Line 400-1. It seems that the promoters are more important than the genes for regulation of prodigiosin production.

21. Discussion – “data not shown” should not be included in the discussion section – this new data should be either moved to the results and shown or kept for a subsequent study. Line 410-414 and 442-445.

22. Figure 1C. The difference between R and WR is not clear in this image. A better picture would be helpful.

23. Figure 1A/B. Rename the Y-axis (left side) to “Surviving CFU/ml” or “Surviving Bacteria (CFU/ml)”

24. not clear whether some of the Y-axes are log scale or linear. Like figure Fig 5C, 7C,

Grammatical issues and writing suggestions:

1. Line 24-5. Please use “Exhibited” rather than exhibiting, add “was” after (named 1912768WR). Cut “in the synthesis of the red pigmented secondary metabolite” as it is possibly confusing.

2. Line 29. Replace “disclosed that RpoS was pleiotropic” with “had pleiotropic effects”

3. Line 32. Replace colon with period.

4. Line 35. add “be” after “might” – also consider that the pig promoter rather than the pig cluster is important.

5. Line 41. cut “with RpoS involving in it”

6. Line Lines 50-55 – giant sentence – please fix.

7. Line 57. Change “secret” to “secrete”

8. Line 70. Cut “simple structured” – it is unnecessary.

9. Lines 74-76. Please fix this sentence.

10. Line 78. Cut “Except for these studies”

11. Line 86. Replace “these” with “this”

12. Line 88. Spelling of regulatory.

13. Line 139. Spelling of queried.

14. Line 146. Spelling of S. marcescens.

15. Line 165. Change “were” to “was”

16. Line 181-2. Remove “detection”, and add “of Slater, et al.” after “method”

17. Lines 193-4. Please fix the sentence.

18. Line 212. Change “was” to “were”

19. Please change “at almost the 16th day” to “sound day 16”.

20. Lines 283-287. Many issues. Add period after type I. Change “besides” to “since or because”, replace “and” on line 284 with a coma. “regulating to regulate”.. change “be the reason causing the loss” to “underly / lead to the observed”. Line 286 add “were” after “that”. Add “for strains WR and W respectively” after “3”.

21. Line 323. Change “impacts” to “effects”

22. Line 342. Change “resulting” with “which is consistent with”

23. Line 344-5. Change “resistant” to “resistance” and cut “ability”

24. Line 366 and 371. Spelling of “membrane permeability” and “medium”

25. Line 373. Space after “colony.”

26. Line 388 and 392. Spelling of “production” and “species”

27. Line 413-420 – add “be” after might on 413 and after may on line 414. Line 415 change rpoS to RpoS, line 418 change “researches are” to “research is”. Line 420, spelling of “strains”.

28. Line 443. Replace “an” with “the”

29. Line 467-8. Cut “in our study”

30. Figure 4B. Y-axis – should be “pigA” rather than PigA.

Reviewer #2: Manuscript Number:

PONE-D-20-03043

Manuscript Title:

Regulation of RpoS in production of prodigiosin and trade-off during prolonged stationary phase in Serratia marcescens 1912768R

This manuscript addresses the problem of loss of prodigiosin production in stationary-phase cultures of Serratia marcescens. Prodigiosin-defective mutants of S. marcescens strain 1912768R were isolated from long-term stationary-phase cultures. Strain 1912768WR was partially deficient in prodigiosin production, while strain 1912768W was entirely deficient in prodigiosin production. The genomes of these two mutants (1912768WR and 1912768W) were sequenced and compared to parent strain 1912768R. Genome analysis revealed that 1912768WR and 1912768W carried mutations in the rpoS gene. An rpoS deletion mutant of parent strain 1912768R was constructed (strain �rpoS) and used in subsequent experiments. Complementation studies were also performed using a plasmid-borne copy of “wild-type” rpoS from parent strain 1912768R. The deletion mutant was defective in prodigiosin production and this phenotype was restored by the complementation plasmid. The authors also show that transcription of the pigment operon (pigA transcript) was downregulated in the deletion mutant. In addition, the authors demonstrate that many other phenotypes were affected by the rpoS deletion including biofilm formation, motility, cellular arrangement, the production of extracellular enzymes, the production of siderophores, and the transcription of genes involved in flagella biosynthesis. The authors also show that the rpoS deletion caused a decrease in resistance to several forms of stress and a general increase in membrane permeability, including changes in OmpC and OmpF levels, and taken together, suggesting a role in the “SPANC balance” of S. marcescens. Lastly, the rpoS deletion strain outcompeted the parent strain in growth studies in both rich and minimal media. The authors conclude that the loss of RpoS is the primary reason for the loss of prodigiosin production in strains 1912768WR and 1912768W.

The data regarding rpoS mutations and their effects on stress resistance, growth, and membrane permeability are not very surprising, as RpoS is a well-known stress-response regulator in the Enterobacteriaceae. The role of RpoS in the stationary-phase-inducible, general-stress-responses of Escherichia coli and Salmonella enterica serovar Typhimurium have been well documented in recent decades, and the authors are missing some important references in this research field. However, the data does provide evidence that prodigiosin is RpoS-regulated in Serratia marcescens. The other data presented in this manuscript provide evidence of an extensive RpoS-regulon in S. marcescens, but how this is relevant to prodigiosin production is less clear. Although the authors show that RpoS-regulated functions are involved in various types of stress resistance and in growth rate, evidence that prodigiosin production is somehow directly connected to the SPANC balance in Serratia marcescens is again unclear. Is a potential connection between prodigiosin and other RpoS-regulated functions being inferred? Does less prodigiosin biosynthesis result in less stress resistance and greater nutritional competence? Perhaps the manuscript should focus on the RpoS regulon of Serratia marcescens instead of just focusing on prodigiosin production. In other words, a different manuscript title might solve some of these issues. Overall, it seems that the study of prodigiosin-deficient mutants led to the discovery of an RpoS regulon in this organism.

Furthermore, the authors begin to discuss possible differences in the RpoS regulation of pig loci (prodigiosin biosynthesis operons) in different S. marcescens strains. This is an interesting point of discussion, but is not explained enough in the manuscript. An additional figure illustrating the potential differences in pig regulation could be helpful. In fact, any potential differences between the RpoS-regulated functions of this organism (1912768R) and those of other S. marcescens strains, or closely related species, should be emphasized more in the body of the manuscript.

The experimental strategies and methods appear to be scientifically sound, and there was sufficient statistical analysis of the data. But again, the authors’ findings with S. marcescens were somewhat predictable based on what is already known about the RpoS regulons of closely related species.

There are extensive English language issues throughout the manuscript. One result of these language problems is that the actual experimental data is not always described sufficiently in a clear manner. Some important experimental results are only described in a couple of sentences, and their impact is lost in the overall body of the manuscript. There are also several typographic errors throughout the manuscript, a few examples of which are listed below:

Line 86: “theses” should be “these”

Line 366: “menbrane” should read “membrane”

Line 388: “produciton” should read “production”

Line 392: “spicies” should be “species”

6. PLOS authors have the option to publish the peer review history of their article (what does this mean?). If published, this will include your full peer review and any attached files.

Reviewer #1: No

Reviewer #2: No

---

## [Author Response · Author response to Decision Letter 0]

4 Apr 2020

Dear Editors and Reviewers:

On behalf of my co-authors, we thank you very much for giving us an opportunity to revise our manuscript. We appreciate the editor’s and reviewers’ positive and constructive comments and suggestions on our manuscript entitled “Regulation of RpoS in production of prodigiosin and trade-off during prolonged stationary phase in Serratia marcescens 1912768R” (PONE-D-20-03043). Those comments are all valuable and very helpful for revising and improving our paper, as well as the important guiding significance to our researches. We have studied the comments carefully and made corrections which we hope meet with approval. Revised portion are marked in red in “Revised Manuscript with Track Changes”. 

Academic Editor：

Thanks to academic editor for the positive comments and suggestions on the requirements of PLOS ONE, we have revised the paper accordingly and thanks again for the comments and suggestions.

Response: Thanks for the comments and suggestions. we have revised the paper accordingly following the PLOS ONE style templates. Also, we have used editing service of American Journal Experts (https://www.aje.cn/) to polish the English language of our manuscript (Certificate Verification Key: 7CE0-5AFC-7E2A-0AE2-36FD). We hope the revised manuscript meets with the requirements of PLOS ONE's style requirements.

2. PLOS requires an ORCID ID for the corresponding author in Editorial Manager on papers submitted after December 6th, 2016. Please ensure that you have an ORCID iD and that it is validated in Editorial Manager. To do this, go to ‘Update my Information’ (in the upper left-hand corner of the main menu), and click on the Fetch/Validate link next to the ORCID field. This will take you to the ORCID site and allow you to create a new iD or authenticate a pre-existing iD in Editorial Manager. Please see the following video for instructions on linking an ORCID iD to your Editorial Manager account: https://www.youtube.com/watch?v=_xcclfuvtxQ

Response: Special thanks for the positive and constructive comments. We have updated the information and authenticated the pre-existing ID (0000-0002-6216-548X).

Response: Thank academic editor for the positive and constructive comments. We will not change the “Data Availability statement”, and we will provide the relevant accession numbers of the genomes at acceptance.

Response: Special thanks for the positive and constructive comments. We have revised the paper accordingly and added the data that is needed in the research; we also removed the phrase that refers to the data, which is not a core part of the research and revised the Discussion part carefully.

5. Please amend your list of authors on the manuscript to ensure that each author is linked to an affiliation. Authors’ affiliations should reflect the institution where the work was done (if authors moved subsequently, you can also list the new affiliation stating “current affiliation:….” as necessary).

Response: Thank you for your positive and constructive comments. We have revised the Authors’ affiliations with “Microbiology and Metabolic Engineering of Key Laboratory of Sichuan Province, College of Life Science, Sichuan University, Chengdu, P.R. China.”

Response: Thanks for the positive comments and suggestions. We have used the PACE to make ensure that the figures of our manuscript meet PLOS requirements.

Replies to Reviewer #1:

Thanks to reviewer#1 for the positive comments on the article and the figures indicating that it could be a useful and interesting paper.

Criticisms:

1. The major issue are the numerous writing issues (grammar and spelling) that weaken the manuscript. I have listed many below, but there are too many to list.

Response: Special thanks for your careful and patient review of the manuscript. We have revised these these wrong spellings and sentences carefully along with the rest of the paper to read more smoothly and be easier to understand. We have used editing service of American Journal Experts (https://www.aje.cn/) to polish the English language of our manuscript (Certificate Verification Key: 7CE0-5AFC-7E2A-0AE2-36FD).

2. Line 21. Please change “which is” to “could be” – this is important because the results of this study suggest that loss of pigmentation will be a problem for industry, but this has not been reported as suggested by the current phrase.

Response: Thank you for your positive and constructive comments. We have changed “which is” to “could be”. This is really important as the problem in industry has not been reported so far.

3. Line 97. inconsistent with line 238. LB should be Lysogeny Broth, but Luria-Bertani is also accepted, but please be accepted.

Response: It should be “Lysogeny Broth”，the article has been revised accordingly，we thank reviewer for the constructive comments and suggestions.

4. Table 1. Inconsistent description of DH5α and S17-1–please add the genotype for S17-1.

Response: Thank reviewer#1 for the positive and constructive comments. we have added the genotype for S17-1.

5. Table 1. Cut “widely used” for two plasmids, also replace knockout vector with “allelic replacement vector”

Response: Thanks for the positive and constructive comments. We have cut “widely used” for the two plasmid and replaced “knockout vector” with “allelic replacement vector”.

6. Line 148-151. Please be more specific about the mutant allele. For example, what base pairs are left.

Response: Special thanks for your your positive and constructive comments. We have pointed out the mutant allele in detail and the article has been revised accordingly.

7. Line 158-9. Replace “double-crossover recombination” with “second recombination”. Double cross-over is incorrect in this allelic replacement approach.

Response: Special thanks for the corrections. We have corrected the wrong description of the allelic replacement approach and we have replaced “double-crossover recombination” with “second recombination”.

8. Methods line 183. Please describe what is meant by “treated chitin” and/or add a reference.

Response: Special thanks for your your positive and constructive comments. The “treated chitin” is confused and we have replaced “treated chitin” with “colloidal chitin”. We have added the reference of preparation of colloidal chitin.

9. Line 185. What is meant by “skim sheep blood”?

Response: “skim sheep blood” is also confused and have been replaced by “sheep erythrocytes”. Thanks for the positive comments and suggestions.

10. Line 215-20. Please describe whether DNase was used in RNA preparation.

Reponse: Thank reviewer #1 for the positive and constructive comments. It is important to describe whether DNase was used in RNA extraction for the untreated RNA have a great influence on the qRT-PCR results. In this study, DNase treatment was performed with Cwbio DNase I at 28°C for 15 min prior to the concentration step and the article has been revised accordingly.

11. Line 234-5. Suggest changing the title to “Selective advantage for non-pigmented mutants during prolonged stationary phase”. Line 236. Change “cause” to “enrich for”.

Response: Special thanks for your your positive and constructive comments. We have changed the title to “Selective advantage for non-pigmented mutants during prolonged stationary phase” and which we do think is more logical. We have replaced “cause” with “enrich for” and it is more accurate.

12. Figure 2. Should say “intergenic region” rather than “intergenic gene”

Response: It should be “intergenic region” and the annotation of “intergenic gene” is wrong. We have changed “intergenic gene” to “intergenic region” in Figure 2 and the article has been revised accordingly. Thanks for the correction.

13. Line 289 – change “type to types”. Moreover, give some information on the amino acid identity of the RpoS protein.

Response: Special thanks for the corrections. We have changed “type” to “types”. The sequences of RpoS are highly conserved with the identity 99.70% (Serratia sp. YD25), 97.89% (Serratia plymuthica AS9), 91.87% (Serratia sp. ATCC 39006) and 94.58% (Escherichia coli K12). We have revised the article accordingly.

14. Line 308 – consider adding the fold-change. It is hard to determine this looking at the linear scale of figure 4B.

Response: Special thanks for Reviewer#1 of the positive and constructive comments. We have changed the linear scale of figure 4B to fold change of 1912768R to ΔrpoS. This modification does make the results more readable.

15. Line 310. Consider adding “indicating that prodigiosin differences were not due to growth”

Response: Special thanks for the positive and constructive comments. The results did suggest that the prodigiosin differences were not due to growth. We have added “indicating that prodigiosin differences were not due to growth” and this is really helpful to describe the impact of the results.

16. Line 320. It was very good using the same promoter for each rpoS gene. Please indicate that all genes were expressed from the rpoS promoter from 1912768R.

Response: Thank you for your positive and constructive comments. We have revised the paper and indicated that the rpoS promoter from 1912768R was synthesized and cloned with the rpoS from different strains to the vector.

17. Line 339-340. “increased by several orders of magnitude” – this suggests that the Y-axis of Figure 5C was a log10 scale – please clarify whether it is log10 or linear.

Response: Thank you for the correction. We are sorry that we used the wrong expression and this caused the misunderstanding. We have changed “increased by several orders of magnitude” to “increased by several times”.

18. Line 359-365. The loss of permeability using these assays is not well described. Are these well-established assays for permeability? I would suggest adding references for these.

Response: Special thanks for your positive and constructive comments. It is important to add the detailed description of these assays, which will increase the credibility of the conclusion. An indirect test for outer membrane permeability is susceptibility to SDS and Cm, as outer membrane exclusion is an important component of protection against these compounds [1-4]. We have made the changes in the paper accordingly and added the references.

19. Line 363-5. Is there any information on the predicted pore size of OmpC and OmpF?

Response: Special thanks for your positive and constructive comments. The information of OmpC and OmpF was limited in the article [4, 5]. OmpC channels are more restrictive than OmpF channels. Bacteria containing only OmpC are more resistant to toxic hydrophilic agents than those containing only OmpF while the transcription of ompF increased when suffering glucose limitation which improved permeability for substrates at low concentrations [1]. We have added the information on the the predicted pore size of OmpC and OmpF in the paper accordingly and added the references.

20. Line 400-1. It seems that the promoters are more important than the genes for regulation of prodigiosin production.

Response: Thank you for your positive and constructive comments. Complementation of RpoS from different strains with the native promoter of rpoS from strain 1912768R restored pigmentation and it seems that the promoters are more important than the genes for regulation of prodigiosin production. We have made the changes in the paper accordingly.

21. Discussion – “data not shown” should not be included in the discussion section – this new data should be either moved to the results and shown or kept for a subsequent study. Line 410-414 and 442-445.

Response: Special thanks for Reviewer#1 of the positive and constructive comments. Line 410-414, the results of transcriptional level of luxS in ΔrpoS mutant compared to the WT strain 1912768R have been described in S3 Fig; Line 442-445, the results is incomplete and still need to be studied further, so we have removed the phrase that refers to these data.

22. Figure 1C. The difference between R and WR is not clear in this image. A better picture would be helpful.

23. Figure 1A/B. Rename the Y-axis (left side) to “Surviving CFU/ml” or “Surviving Bacteria (CFU/ml)”

Response: Special thanks for your positive and constructive comments. The difference of the color between 1912768R and 1912768WR is difficult to distinguish and we have changed the picture to make them easier to distinguish. We have renamed Y-axis (left side) to “Surviving CFU/ml”.

24. not clear whether some of the Y-axes are log scale or linear. Like figure Fig 5C, 7C,

Response: Thank you for the correction. We are sorry that we used the wrong expression and this caused the misunderstanding. We have changed “increased by several orders of magnitude” to “increased by several times” in the description of Fig 5C. Fig 7C is the relative transcription of corresponding genes in ΔrpoS compared to WT and the Y-axe is linear. We have revised the paper accordingly.

Grammatical issues and writing suggestions:

1. Line 24-5. Please use “Exhibited” rather than exhibiting, add “was” after (named 1912768WR). Cut “in the synthesis of the red pigmented secondary metabolite” as it is possibly confusing.

Response: Special thanks for your careful and patient review. We have replaced “Exhibited” with “exhibiting”, added “was” after (named 1912768WR), and cut the “in the synthesis of the red pigmented secondary metabolite”.

2. Line 29. Replace “disclosed that RpoS was pleiotropic” with “had pleiotropic effects”

Response: Thank you for your careful and patient review. We have replaced “disclosed that RpoS was pleiotropic” with “had pleiotropic effects” which is more accurate.

3. Line 32. Replace colon with period.

Response: Thank you for the correction. However, we could not find the word “colon” in Line 32. Once again, thank you very much for your comments and suggestions.

4. Line 35. add “be” after “might” – also consider that the pig promoter rather than the pig cluster is important.

Response: Thank you for the positive comments and suggestions. We have added “be”after “might” and revised the paper as recommended.

5. Line 41. cut “with RpoS involving in it”

Response: We have cut “with RpoS involving in it” which is not necessary. Thanks for the the positive comments and suggestions.

6. Line Lines 50-55 – giant sentence – please fix.

Response: Thanks for the positive comments and suggestion. We have fixed the sentence and revised the paper correspondingly.

7. Line 57. Change “secret” to “secrete”

Response: It should be “secrete” and we have replaced “secret” with “secrete”. Thanks for the correction.

8. Line 70. Cut “simple structured” – it is unnecessary.

Response: Thanks for the positive comments and suggestion. We have cut the “simple structured”, which is not necessary.

9. Lines 74-76. Please fix this sentence.

Response: Thank reviewer#1 for the positive comments and suggestion. We have fixed the sentence and revised the paper correspondingly.

10. Line 78. Cut “Except for these studies”

Response: Thanks for the positive comments and suggestions. We have cut “Except for these studies” and revised the paper correspondingly.

11. Line 86. Replace “these” with “this”

Response: It should be “this”, and we have replaced “these” with “this”. Thanks for the correction.

12. Line 88. Spelling of regulatory.

Response: It should be “regulatory”. We have replaced “regularoty” with “regulatory” and checked the spelling of the whole manuscript. Thanks for the careful and patient review of the paper.

13. Line 139. Spelling of queried.

Response: Thanks for the correction. We have revised the word accordingly.

14. Line 146. Spelling of S. marcescens.

Response: It should be “S. marcescens”. We have checked the spelling of the whole manuscript. Thanks for the careful and patient review of the paper.

15. Line 165. Change “were” to “was”

Response: It should be “was” and we have changed “were” to “was”. Thanks for the correction.

16. Line 181-2. Remove “detection”, and add “of Slater et al.” after “method”

Response: Thanks for the positive comments and suggestions. We have removed detection and added “of Slater et al.” after “method”. This makes the sentence more concise and easier to understand.

17. Lines 193-4. Please fix the sentence.

Response: Thank you for the corrections. We have fixed the giant sentence to make it more concise and easier to understand, and we have revised in the paper accordingly.

18. Line 212. Change “was” to “were” 

Response: It should be “were” and we have replaced “was” with “were”. Thank reviewer#1 for the correction.

19. Please change “at almost the 16th day” to “sound day 16”.

Response: Thanks for the positive comments and suggestions. We have changed “at almost the 16th day” to “sound day 16”.

20. Lines 283-287. Many issues. Add period after type I. Change “besides” to “since or because”, replace “and” on line 284 with a coma. “regulating to regulate”. change “be the reason causing the loss” to “underly / lead to the observed”. Line 286 add “were” after “that”. Add “for strains WR and W respectively” after “3”.

Response: Thanks for the corrections. We have replaced “besides” with “since”. We have replaced “and” on line 284 with a coma. We have changed “regulating” to “regulate”. We have changed “be the reason causing the loss” to “lead to the observed”. We have added “were” after “that” and added “for strains WR and W respectively” after “3”.

21. Line 323. Change “impacts” to “effects”

Response: Thanks for the correction. It should be “effects” and we have changed “impacts” to “effects” which is more concise.

22. Line 342. Change “resulting” with “which is consistent with”

Response: Thank you for the positive suggestion. We have changed “resulting” with “which is consistent with”.

23. Line 344-5. Change “resistant” to “resistance” and cut “ability”

Response: It should be “resistance” and we have changed “resistant” to “resistance”; moreover, we have cut “ability”. Thanks for the carefully review of our manuscript which is very helpful.

24. Line 366 and 371. Spelling of “membrane permeability” and “medium”

Response: We have corrected the spelling mistake of the word “membrane permeability” and “medium” and thanks for the corrections.

25. Line 373. Space after “colony.”

Response: Special thanks for your careful and patient review of the manuscript. We are really sorry that we made many spelling mistakes in this paper and this would cause the misunderstanding. We have revised the mistakes correspondingly and checked the whole paper carefully.

26. Line 388 and 392. Spelling of “production” and “species”

Response: Thanks for the correction. We have corrected the spelling of the words “production” and “species”.

27. Line 413-420 – add “be” after might on 413 and after may on line 414. Line 415 change rpoS to RpoS, line 418 change “researches are” to “research is”. Line 420, spelling of “strains”.

Response: Special thanks for your careful and patient review. We have checked Lines 413-420 carefully and revived as recommended accordingly. We have added “be” after “might” and “may”. It should be RpoS and we have changed “rpoS” to “RpoS”. We have changed “researches are” to “research is”. We have correct the spelling of the word “strains”.

28. Line 443. Replace “an” with “the”

Response: We have changed “an” to “the” and thanks again for the correction.

29. Line 467-8. Cut “in our study”

Response: We have cut the words “in our study” and thanks for the corrections.

30. Figure 4B. Y-axis – should be “pigA” rather than PigA.

Response: Special thanks for your positive and constructive comments. It should be pigA, and we have changed “PigA” rather than pigA.

Replies to Reviewer#2:

1. The data regarding rpoS mutations and their effects on stress resistance, growth, and membrane permeability are not very surprising, as RpoS is a well-known stress-response regulator in the Enterobacteriaceae. The role of RpoS in the stationary-phase-inducible, general-stress-responses of Escherichia coli and Salmonella enterica serovar Typhimurium have been well documented in recent decades, and the authors are missing some important references in this research field. 

Response: We thank reviewer#2 for their positive and constructive comments. The stationary phase sigma factor RpoS is the master regulator of the general stress response (GSR) in E. coli [6]. RpoS is known to play an important role in adaptation of bacteria to general stresses such as carbon starvation, high temperature and oxidative stress [7-9]. However, the regulatory role of RpoS in environmental stress is diverse in different strains [10-12]. In E. coli, RpoS is positively related to the oxidative resistance, while in Borrelia burgdorferi, RpoS is negatively related to oxidative resistance [10, 13, 14]. In Burkholderia pseudomallei and Pseudomonas aeruginosa, role of RpoS in regulating resistance of osmotic stress is also different [10, 15, 16]. In different E. coli strains, the capability of resistance to the same stress is different in different rpoS mutant strains [17]. Therefore, although RpoS is a well-known stress-response regulator, it seems that the regulatory role of RpoS in stress resistance is not all the same in different species. At present, the regulatory role of RpoS in stress resistance in S. marcescens has not been reported and the specific role of RpoS in S. marcescens is still unclear. So, it seems that the work is not without merit. Actually, we do have found some interesting points in this study.

The role of RpoS in the stationary-phase-inducible, general-stress-responses of E. coli and S. enterica serovar Typhimurium have been well documented in recent decades and we have added the corresponding references to illustrate the research progress of RpoS in GSR of Escherichia coli and Salmonella enterica serovar Typhimurium. We have revised the “Introduction” and “Results” of the paper accordingly. Thanks again for the positive and constructive comments.

2. However, the data does provide evidence that prodigiosin is RpoS-regulated in Serratia marcescens. The other data presented in this manuscript provide evidence of an extensive RpoS-regulon in S. marcescens, but how this is relevant to prodigiosin production is less clear. 

Response: Special thanks to reviewers for their positive comments on the conclusions of the article. The regulatory genes of prodigiosin production seem to be pleiotropic. The LysR-type transcriptional regulator, MetR, is a positively regulator in regulation of prodigiosin biosynthesis and it is also pleiotropic in regulation of methionine biosynthesis, cell motility, H2O2 tolerance, heat tolerance, and exopolysaccharide synthesis in Serratia marcescens [18]; An IgaA/UmoB family protein from Serratia marcescens is a positive regulator in prodigiosin regulation and it also regulates motility, capsular polysaccharide biosynthesis, and serratamolide production [19]; The positive regulator of prodigiosin production, RbsR, is a pleiotropic regulator of motility, virulence, siderophore and antibiotic production, gas vesicle morphogenesis and flotation in Serratia sp. ATCC 39006. Therefore, it seems that regulation of prodigiosin production is relevant to other phenotypes. However, the relationship between prodigiosin production and other phenotypes has not been reported. In this study, we do not have the direct evidences to prove the specific relationship between prodigiosin production and other diverse phenotypes, but we have proved that prodigiosin production and other phenotypes are regulated by RpoS and it seems that the extensive RpoS-regulon is relevant to prodigiosin production. Thanks again for your positive and constructive comments and we will continue to explore their specific relationship in future researches.

3. Although the authors show that RpoS-regulated functions are involved in various types of stress resistance and in growth rate, evidence that prodigiosin production is somehow directly connected to the SPANC balance in Serratia marcescens is again unclear. Is a potential connection between prodigiosin and other RpoS-regulated functions being inferred? Does less prodigiosin biosynthesis result in less stress resistance and greater nutritional competence? Perhaps the manuscript should focus on the RpoS regulon of Serratia marcescens instead of just focusing on prodigiosin production. In other words, a different manuscript title might solve some of these issues. Overall, it seems that the study of prodigiosin-deficient mutants led to the discovery of an RpoS regulon in this organism.

Response：Thank reviewer#2 for the positive and constructive comments. The physiological role of bacterial prodigiosin in vivo remains undefined [20]. A number of reports have suggested potential functions that may provide an advantage in resistance to other strains (biotic stress) and challenging natural environment (abiotic stress) [21, 22]. In this study, the ΔrpoS strain was unable to produce prodigiosin and the resistance of ΔrpoS strain to different stresses decreased compared to the pigment WT strain at the same time; in addition, the ΔrpoS strain outcompeted the pigment WT strain. Therefore, this results indirectly proved that less prodigiosin biosynthesis results in less stress resistance and greater nutritional competence. Not only RpoS but also the prodigiosin is relevant to the SPANC balance. However, the poor English style and grammar of this manuscript makes some sections of the manuscript very difficult to read and the manuscript title is not appropriate. It is the study of prodigiosin-deficient mutants leading to the discovery of an RpoS regulon in this organism. 

We have revised the whole manuscript carefully and changed the title with “RpoS is a pleiotropic regulator of motility, biofilm formation, exoenzymes, siderophore and prodigiosin production, and trade-off during prolonged stationary phase in Serratia marcescens”. 

4. Furthermore, the authors begin to discuss possible differences in the RpoS regulation of pig loci (prodigiosin biosynthesis operons) in different S. marcescens strains. This is an interesting point of discussion, but is not explained enough in the manuscript. An additional figure illustrating the potential differences in pig regulation could be helpful. In fact, any potential differences between the RpoS-regulated functions of this organism (1912768R) and those of other S. marcescens strains, or closely related species, should be emphasized more in the body of the manuscript.

Response: Special thanks for your your positive and constructive comments. We have added additional figure to illustrate the potential difference in pig regulation (Fig 8). As the regulatory pathway of RpoS in prodigiosin production is unclear, it is hard to fully clarify the reason why the regulatory role of RpoS is different in Serratia marcescens 1912768R and strain Serratia sp. ATCC 39006. We have emphasized the potential differences between the RpoS-regulated functions of strain 1912768R and those of ATCC 39006 and revised the paper accordingly.

5. The experimental strategies and methods appear to be scientifically sound, and there was sufficient statistical analysis of the data. But again, the authors’ findings with S. marcescens were somewhat predictable based on what is already known about the RpoS regulons of closely related species.

Response: Special thanks to reviewers for their positive comments on the experimental strategies and methods of the article. RpoS is a master regulator of different phenotypes and the RpoS regulons have been studied in many species due to the important role in regulation of exoenzymes secretion, stress resistance and secondary metabolites biosynthesis. 

In Vibrio anguillarum, RpoS positively regulates the secretion of extracellular lipase and proteinase, and hemolysis ability [14]; on the contrary, secretion of extracellular lipase and proteinase increased in rpoS mutant strain of Pseudomonas chlororaphis PA23 [23, 24]. In Borrelia burgdorferi and Vibrio cholerae, the secretion of chitinase is dependent on RpoS, while in S. marcescens 1912768R, secretion of chitinase increased in rpoS mutant strains. In addition, the regulatory role of RpoS in environmental stress is diverse in different strains (please see the answer to question 1). In Pseudomonas species, the regulatory effect of RpoS on secondary metabolites is diverse or even opposite in some species [25]; it was until we sequenced and deleted rpoS in S. marcescens 1912768R that we prove RpoS is a positive regulator in prodigiosin biosynthesis and its regulatory role in prodigiosin production is opposite to that in Serratia sp. ATCC 39006 [26]. Therefore, the regulatory role of RpoS is diverse in different strains. Although RpoS have been proved to be a regulator in exoenzymes secretion and stress resistance, the specific role of RpoS in regulation of exoenzymes secretion and stress resistance is still unknown in S. marcescens. Some results of this paper may be predictable, but we do have some new findings reported. We tried our best to improve the manuscript and make revisions to make the points clearer. Once again, thank you very much for your comments and suggestions.

6. There are extensive English language issues throughout the manuscript. One result of these language problems is that the actual experimental data is not always described sufficiently in a clear manner. Some important experimental results are only described in a couple of sentences, and their impact is lost in the overall body of the manuscript. There are also several typographic errors throughout the manuscript, a few examples of which are listed below:

Line 86: “theses” should be “these” 

Line 366: “menbrane” should read “membrane” 

Line 388: “produciton” should read “production” 

Line 392: “spicies” should be “species”

Response: Special thanks for your careful and patient review. The results of the manuscript have been revised and we have used editing service of American Journal Experts (https://www.aje.cn/) to polish the English language of our manuscript. It should be “these” and we have replaced “theses” with “these”. We have replaced “menbrane” with “membrane”. The wrong word “produciton” have been replaced with “production”. It should be “species” and we have removed the wrong word “spicies”. 

We tried our best to improve the manuscript and make revisions following the reviewers’ comments in the manuscript. These changes will not influence the content or framework of the paper. We earnestly appreciate the the editor’s and reviewers’ thorough work, and hope that the revision will meet with their approval. 

Once again, thank you very much for your comments and suggestions.

Reference

1. Liu X, Ferenci T. An analysis of multifactorial influences on the transcriptional control of ompF and ompC porin expression under nutrient limitation. Microbiology. 147(11):2981-9.

2. Nikaido, H. Molecular Basis of Bacterial Outer Membrane Permeability Revisited. 2003.

3. Pagès J-M, James CE, Winterhalter M. The porin and the permeating antibiotic: a selective diffusion barrier in Gram-negative bacteria. 6(12):893-903.

4. Delcour AH. Outer membrane permeability and antibiotic resistance. Biochimica Et Biophysica Acta Proteins & Proteomics. 1794(5):808-16.

5. Liu XQ, Ferenci T. Regulation of Porin-Mediated Outer Membrane Permeability by Nutrient Limitation in Escherichia coli. Journal of Bacteriology. 1998;180(15):3917-22.

6. Hengge R. Proteolysis of sigmaS (RpoS) and the general stress response in Escherichia coli. Res Microbiol. 2009;160(9):667-76. doi: 10.1016/j.resmic.2009.08.014. PubMed PMID: 19765651.

7. Battesti A, Majdalani N, Gottesman S. The RpoS-mediated general stress response in Escherichia coli. Annu Rev Microbiol. 2011;65:189-213. doi: 10.1146/annurev-micro-090110-102946. PubMed PMID: 21639793.

8. Landini P, Egli T, Wolf J, Lacour S. sigmaS, a major player in the response to environmental stresses in Escherichia coli: role, regulation and mechanisms of promoter recognition. Environ Microbiol Rep. 2014;6(1):1-13. doi: 10.1111/1758-2229.12112. PubMed PMID: 24596257.

9. Somorin Y, Bouchard G, Gallagher J, Abram F, Brennan F, O'Byrne C. Roles for RpoS in survival of Escherichia coli during protozoan predation and in reduced moisture conditions highlight its importance in soil environments. FEMS Microbiol Lett. 2017;364(19). doi: 10.1093/femsle/fnx198. PubMed PMID: 28967947.

10. Dong T, Schellhorn HE. Role of RpoS in virulence of pathogens. Infect Immun. 2010;78(3):887-97. doi: 10.1128/IAI.00882-09. PubMed PMID: 19948835; PubMed Central PMCID: PMCPMC2825926.

11. Schellhorn, E H. Elucidating the function of the RpoS regulon. Future Microbiology. 9(4):497-507.

12. Liu X, Ji L, Wang X, Li J, Zhu J, Sun A. Role of RpoS in stress resistance, quorum sensing and spoilage potential of Pseudomonas fluorescens. Int J Food Microbiol. 2018;270:31-8. doi: 10.1016/j.ijfoodmicro.2018.02.011. PubMed PMID: 29471265.

13. Caimano MJ, Eggers CH, Hazlett KRO, Radolf JD. RpoS Is Not Central to the General Stress Response in Borrelia burgdorferi but Does Control Expression of One or More Essential Virulence Determinants. Infection & Immunity. 72(11):6433-45.

14. Elias AF, Bono JL, Carroll JA, Stewart P, Tilly K, Rosa P. Altered Stationary-Phase Response in a Borrelia burgdorferi rpoS Mutant. Journal of Bacteriology. 182(10):2909-18.

15. Sonnleitner E, Hagens S, Rosenau F, Wilhelm S, Habel A, J?ger K-E, et al. Reduced virulence of a hfq mutant of Pseudomonas aeruginosa O1. Microb Pathog. 35(5):0-228.

16. Subsin B, Thomas MS, Katzenmeier G, Shaw JG, Kunakorn M. Role of the Stationary Growth Phase Sigma Factor RpoS of Burkholderia pseudomallei in Response to Physiological Stress Conditions. Journal of Bacteriology. 2004;185(23):7008-14.

17. Mata G, Ferreira GM, Spira B. RpoS role in virulence and fitness in enteropathogenic Escherichia coli. PLoS One. 2017;12(6):e0180381. doi: 10.1371/journal.pone.0180381. PubMed PMID: 28662183; PubMed Central PMCID: PMCPMC5491219.

18. Pan X, Sun C, Tang M, You J, Osire T, Zhao Y, et al. LysR-Type Transcriptional Regulator MetR Controls Prodigiosin Production, Methionine Biosynthesis, Cell Motility, H2O2 Tolerance, Heat Tolerance, and Exopolysaccharide Synthesis in Serratia marcescens. Appl Environ Microbiol. 2020;86(4). doi: 10.1128/AEM.02241-19. PubMed PMID: 31791952; PubMed Central PMCID: PMCPMC6997736.

19. Stella NA, Brothers KM, Callaghan JD, Passerini AM, Sigindere C, Hill PJ, et al. An IgaA/UmoB Family Protein from Serratia marcescens Regulates Motility, Capsular Polysaccharide Biosynthesis, and Secondary Metabolite Production. Appl Environ Microbiol. 2018;84(6). doi: 10.1128/AEM.02575-17. PubMed PMID: 29305504; PubMed Central PMCID: PMCPMC5835751.

20. Yip CH, Yarkoni O, Ajioka J, Wan KL, Nathan S. Recent advancements in high-level synthesis of the promising clinical drug, prodigiosin. Appl Microbiol Biotechnol. 2019;103(4):1667-80. doi: 10.1007/s00253-018-09611-z. PubMed PMID: 30637495.

21. Das A, Bhattacharya S, Gulani C. Assessment of Process Parameters Influencing the Enhanced Production of Prodigiosin from Serratia marcescensand Evaluation of Its Antimicrobial, Antioxidant and Dyeing Potentials. Malaysian Journal of Microbiology. 2012. doi: 10.21161/mjm.03612.

22. Stankovic N, Senerovic L, Ilic-Tomic T, Vasiljevic B, Nikodinovic-Runic J. Properties and applications of undecylprodigiosin and other bacterial prodigiosins. Applied Microbiology & Biotechnology. 98(9):3841-58.

23. Selin C, Manuel J, Fernando WGD, de Kievit T. Expression of the Pseudomonas chlororaphis strain PA23 Rsm system is under control of GacA, RpoS, PsrA, quorum sensing and the stringent response. Biological Control. 69:24-33.

24. Poritsanos NJ. Molecular mechanisms involved in secondary metabolite production and biocontrol of Pseudomonas chlororaphis PA23. 2006.

25. Girard G, van Rij ET, Lugtenberg BJ, Bloemberg GV. Regulatory roles of psrA and rpoS in phenazine-1-carboxamide synthesis by Pseudomonas chlororaphis PCL1391. Microbiology. 2006;152(Pt 1):43-58. doi: 10.1099/mic.0.28284-0. PubMed PMID: 16385114.

26. Wilf NM, Salmond GP. The stationary phase sigma factor, RpoS, regulates the production of a carbapenem antibiotic, a bioactive prodigiosin and virulence in the enterobacterial pathogen Serratia sp. ATCC 39006. Microbiology. 2012;158(Pt 3):648-58. doi: 10.1099/mic.0.055780-0. PubMed PMID: 22194349.

---

## [Decision Letter · Decision Letter 1]

17 Apr 2020

RpoS is a pleiotropic regulator of motility, biofilm formation, exoenzymes, siderophore and prodigiosin production, and trade-off during prolonged stationary phase in Serratia marcescens

PONE-D-20-03043R1

Dear Dr. Cao,

We are pleased to inform you that your manuscript has been judged scientifically suitable for publication and will be formally accepted for publication once it complies with all outstanding technical requirements.

With kind regards,

Marie-Joelle Virolle, PhD

Academic Editor

PLOS ONE

Additional Editor Comments (optional):

Reviewers' comments:

Reviewer's Responses to Questions

**Comments to the Author**

1. If the authors have adequately addressed your comments raised in a previous round of review and you feel that this manuscript is now acceptable for publication, you may indicate that here to bypass the “Comments to the Author” section, enter your conflict of interest statement in the “Confidential to Editor” section, and submit your "Accept" recommendation.

Reviewer #1: All comments have been addressed

2. Is the manuscript technically sound, and do the data support the conclusions?

Reviewer #1: Yes

3. Has the statistical analysis been performed appropriately and rigorously? 

Reviewer #1: Yes

4. Have the authors made all data underlying the findings in their manuscript fully available?

Reviewer #1: Yes

5. Is the manuscript presented in an intelligible fashion and written in standard English?

Reviewer #1: Yes

6. Review Comments to the Author

Reviewer #1: The authors adequately addressed my comments and concerns and the manuscript is greatly improved. I think that it is a nice addition to the field.

7. PLOS authors have the option to publish the peer review history of their article (what does this mean?). If published, this will include your full peer review and any attached files.

Reviewer #1: Yes: Robert M. Q. Shanks

---

## [Editor Report · Acceptance letter]

8 May 2020

PONE-D-20-03043R1 

RpoS is a pleiotropic regulator of motility, biofilm formation, exoenzymes, siderophore and prodigiosin production, and trade-off during prolonged stationary phase in *Serratia marcescens*

Dear Dr. Cao:

I am pleased to inform you that your manuscript has been deemed suitable for publication in PLOS ONE. Congratulations! Your manuscript is now with our production department. 

With kind regards,

on behalf of

Dr. Marie-Joelle Virolle 

Academic Editor

PLOS ONE